# Peripheral immune system modulates Purkinje cell degeneration in Niemann–Pick disease type C1

Toru Yasuda[1] , Toru Uchiyama[1], Nobuyuki Watanabe[1], Noriko Ito[2] , Kazuhiko Nakabayashi[2], Hideki Mochizuki[3], Masafumi Onodera[1]

**Niemann–Pick disease type C1 (NPC1) is a fatal lysosomal storage disorder characterized by progressive neuronal degeneration. Its key pathogenic events remain largely unknown. We have, herein, found that neonatal BM–derived cell transplantation can ameliorate Purkinje cell degeneration in NPC1 mice. We subsequently addressed the impact of the peripheral immune system on the neuropathogenesis observed in NPC1 mice. The depletion of mature lymphocytes promoted NPC1 phenotypes, thereby suggesting a neuroprotective effect of lymphocytes. Moreover, the peripheral infusion of CD4-positive cells (specifically, of regulatory T cells) from normal healthy donor ameliorated the cerebellar ataxic phenotype and enhanced the survival of Purkinje cells. Conversely, the depletion of regulatory T cells enhanced the onset of the neurological phenotype. On the other hand, circulating inflammatory monocytes were found to be involved in the progression of Purkinje cell degeneration, whereas the depletion of resident microglia had little effect. Our findings reveal a novel role of the adaptive and the innate immune systems in NPC1 neuropathology.**

## Introduction

Niemann–Pick disease type C (NPC) is an autosomal recessive lysosomal storage disorder (LSD) affecting an estimated 1 in 120,000 newborns worldwide (Vanier, 2010). Mutations in *NPC1* can be found in ~95% of NPC patients, although the gene's product (the NPC1 protein) functions in late endosomes and lysosomes, where it transports unesterified cholesterol to the plasma membrane and to cellular organelles (Kwon et al, 2009; Vanier, 2010; Vance & Karten, 2014). Affected individuals display a neurovisceral accumulation of unesterified cholesterol and several forms of glycosphingolipids (Davidson et al, 2009), along with clinical manifestations ranging from neonatal acute fatality to adult-onset chronic disease associated with neurodegeneration (Patterson et al, 2013). Neurological

symptoms are usually preceded by systemic signs that include cholestatic jaundice during the neonatal period and (hepato) splenomegaly in infancy or childhood. Moreover, neurological symptoms, such as cerebellar ataxia, laughter-induced cataplexy, dystonia, supranuclear gaze palsy, and progressive dementia, are frequently observed in juvenile and adult-onset NPC patients (Vanier, 2010).

There is currently no curative treatment for NPC. Several potential therapies aim at alleviating the neurological symptoms of NPC. Among these, *N*-butyldeoxynojirimycin (also known as miglustat or by the trade name "Zavesca"), an inhibitor of glucosylceramide synthase that has been developed for the treatment of Gaucher disease, can reduce the glycosphingolipid levels and stabilize the neurological symptoms of NPC (Patterson et al, 2007, 2020; Pineda et al, 2018; Sitarska et al, 2021). This is the only drug that is currently approved for use in the treatment of NPC in several countries. However, the treated patients frequently suffer from side effects such as chronic diarrhea, weight decrease, flatulence, and seizures (Wraith et al, 2010; Patterson et al, 2020). Another potential therapeutic agent is 2-hydroxypropyl-$\beta$-cyclodextrin (HP$\beta$CD), a cyclic oligosaccharide derivative and a strong cholesterol solubilizer. HP$\beta$CD has been shown to slow the progression of the disease and to expand the lifespan of NPC model animals (Vance & Karten, 2014; Ishitsuka et al, 2022). However, this compound cannot cross the blood–brain barrier (BBB) and can cause severe hearing loss in mice through its selective cytotoxicity on cochlear outer hair cells (Crumling et al, 2012, 2017). The intrathecal administration of HP$\beta$CD has generated promising results in several clinical trials by suppressing the progression of the disease (Ory et al, 2017; Berry-Kravis et al, 2018; Calias, 2018; Farmer et al, 2019). The removal of the accumulating cholesterol and other lipids from the cerebral cells may be an indispensable prerequisite for the improvement of neurological symptoms, whereas an exploration of key pathogenic mechanisms promoting the neurodegenerative process could contribute to the development of more effective therapies.

Because the biochemical improvement and the reversal of the clinical features of the Hurler syndrome (Hobbs et al, 1981), hematopoietic cell transplantation (HCT) from allogeneic healthy

---

[1]Department of Human Genetics, National Center for Child Health and Development, Tokyo, Japan   [2]Department of Maternal-Fetal Biology, National Center for Child Health and Development, Tokyo, Japan   [3]Department of Neurology, Graduate School of Medicine, Osaka University, Osaka, Japan

Correspondence: yasuda-t@ncchd.go.jp

donors has been adopted for the treatment of several LSDs (Peters & Steward, 2003; Staba et al, 2004; Escolar et al, 2005). The transplanted hematopoietic cells and/or their progenies can provide a stable source of non-defective enzyme in the affected tissues. In fact, the active enzyme excreted from the reconstituted cells can be internalized by host cells, thereby resulting in a correction of the metabolic defect. However, the efficacy of HCT has only been proven in a few selected LSDs so far (Walker & Montell, 2016). Moreover, very few studies have reported a successful improvement of neurological symptoms by HCT (Peters & Steward, 2003). On the other hand, autologous hematopoietic stem cell transplantation (aHSCT) is currently considered as a highly effective treatment for severe autoimmune diseases such as multiple sclerosis (MS) (Giedraitiene et al, 2022; Nabizadeh et al, 2022; Ruder et al, 2022; Willison et al, 2022). Inflammation, demyelination, and neuro-axonal degeneration represent major aspects of the MS neuropathology and are primarily due to the infiltration of the central nervous system (CNS) by autoimmune T and B lymphocytes (Dendrou et al, 2015). In experimental autoimmune encephalomyelitis (EAE), an animal model of MS, the infiltrating monocyte-derived macrophages can also initiate demyelination at disease onset, whereas resident microglia may be involved in debris clearing (Ajami et al, 2011; Yamasaki et al, 2014; Dendrou et al, 2015). aHSCT was designed so as to facilitate the removal of harmful immune cells and to induce an "immune reset" with a newly generated immune repertoire. Immune dysregulation–associated neuroinflammation represents an important neuropathological mechanism and a potential therapeutic target for several neurodegenerative diseases, including Alzheimer disease (AD), Parkinson disease (PD), and amyotrophic lateral sclerosis (ALS) (Appel & Beers, 2019; Tansey et al, 2022; Zang et al, 2022).

In view of the extensive activation of microglia/macrophages in the NPC CNS (Baudry et al, 2003; Platt et al, 2016; Cougnoux et al, 2018a; Colombo et al, 2021), we have, herein, aimed at elucidating the involvement of the resident and the peripheral innate/adaptive immune systems in the neurodegenerative processes of NPC. We used a mouse model of the *NPC1*-associated form of NPC (NPC1; Niemann–Pick disease type C1), that is, characterized by a non-detectable expression of the NPC1 protein and has been widely used as an acutely progressive neurodegenerative model of the disease (Loftus et al, 1997). We have found that the neonatal BM–derived cell transplantation from a healthy donor exerts a neuroprotective effect on cerebellar Purkinje cells that are selectively killed in NPC. Neuronal degeneration was then examined by selectively depleting peripheral lymphocytes, inflammatory monocytes, and resident microglia. Our study reveals a modulation of neuronal degeneration by peripheral adaptive and innate immune systems, and our findings could contribute to the development of new therapies for this devastating neurodegenerative disease.

## Results

### Amelioration of Purkinje cell loss by neonatal BM–derived cell transplantation

We set out to determine the effect of BM-derived cell transplantation and the potential involvement of the peripheral immune

system in the neuropathogenic process of NPC1. We used *Npc1^{nih}* mice (hereafter, designated also as "NPC1 mice" or "npc1−/− mice" when compared with their wild-type npc1+/+ littermates) that recapitulate many aspects of the phenotypes of NPC (Loftus et al, 1997). In an attempt to select a most appropriate timepoint for the determination of neuronal degeneration, we first analyzed a cerebellar ataxic phenotype of *Npc1^{nih}* mice in 4–10-wk-old mice through an accelerating rotarod test. The mutant mice displayed an aggravation of cerebellar ataxia at 39 d of age and at later timepoints when compared with their wild-type littermates and reached a minimal score at around 49 d of age (7-wk-old mice) (Fig 1A). The body weight of the mutant mice was lower when compared with that of the wild-type mice, although it reached its peak and began to diminish when the mice were ~39–49 d of age (~6-7-wk-old) (Fig 1B and C). Subsequently, we performed an immunohistochemical analysis for the calbindin-positive Purkinje cells in the cerebellum, and we counted the number of these cells in every lobule of 5-, 7-, and 10-wk-old mice. The 7-wk-old npc1−/− mice displayed a loss of calbindin-positive Purkinje cells in their anterior lobules, whereas the 10-wk-old npc1−/− mice displayed a loss in all of their lobules except for the lobule X (Fig 1D–J). The survival time of the mutant mice ranged approximately from 10 to 15 wk, with no evidence of a gender-based difference (Fig 1K). We selected the "7 wk" as a timepoint for the analysis of the therapeutic effect of BM cell transplantation. Before the examination, we investigated a change of immune cells in the peripheral blood of 5- and 7-wk-old NPC1 mice. We found that the percentage of CD4-positive cells over CD45-positive cells was decreased, whereas that of CD11b-positive and CD11b–Ly6C double-positive cells over CD45-positive cells was increased in the peripheral blood of both female and male NPC1 mutant mice (Fig 1L, N–P, R, and S). The percentage of CD4–CD25–Foxp3–triple-positive regulatory T cells (Tregs) was unaltered in these mice (Fig 1M and Q).

Neonatal pups (P2–P6) received a single injection with busulfan (40 μg/g) followed by an i.p. injection of BM mononuclear cells that were isolated from Kusabira-Orange (KuO) transgenic (Tg) mice (1–2 × 10^6 cells per pup). When the mice reached 7 wk of age, the CD11b-positive cells of their peripheral blood were analyzed for KuO fluorescence through flow cytometry. A mouse in which more than 10% of its CD11b-positive cells were positive for KuO was designated as a "successfully BM-cell transplanted (BMT)" animal, whereas a mouse in which less than 10% of its CD11b-positive cells were positive for KuO was designated as "negative control (nc)." Although the BMT female and male mice exhibited a non-significant trend for increase in their rotarod scores (Fig 2A and B), their cerebellar Purkinje cells were significantly preserved only in the case of the BMT female mice (Fig 2C–G and C'–F'). The analysis of their peripheral blood revealed an increase of CD4-positive and CD4–CD25 double-positive cells, and a decrease of CD11b-positive cells (Fig 2H–K). The histological analysis of the KuO-positive cells in the cerebellum revealed an invasion of the KuO–CD11b double-positive monocyte-derived macrophages into the perivascular zone of the interlobular space of the cerebellum in both female and male mice (Figs 2M–O and S1A–R). We were not able to identify any KuO-positive cells in the brains of BMT wild-type npc1+/+ mice (Fig 2L). When the BM cell transplantation was performed using npc1−/−_KuO–Tg mice as a donor, we observed an invasion of the KuO–

CD11b double-positive monocyte–derived macrophages into the perivascular zones of the interlobular space, the molecular layer (ML), and the Purkinje cell layer (PCL) of the cerebellum in both female and male mice (Fig S1S–Z). These findings suggest that peripheral immune cells can be involved in the modulation of the neurodegenerative process of NPC1 and have prompted us to examine the effect of the depletion of peripheral lymphocytes or tissue-infiltrating inflammatory monocytes on the survival of cerebellar Purkinje cells. Breaching of the BBB was confirmed histologically by a leakage of peripherally injected Evans blue dye and sulfo-NHS-biotin molecules into the parenchyma of the cerebellum of NPC1 mice (Fig S1A'–H') that could not be detected by a biochemical method in a previous work (Cougnoux et al, 2018a). Decreased immunoreactivities for laminin (Fig S1A''–F'') and collagen type IV (Fig S1G''–L'') were also revealed by the co-immunostaining of CD31-positive blood vessels. We did not find any sex differences in these experiments. These findings suggest that the basement membrane of the cerebellar blood vessels may be damaged in NPC1 mice.

## Depletion of lymphocytes enhanced the neurodegenerative phenotype in NPC1 mice

We generated NPC1 mutant mice that are deficient for mature T and B lymphocytes by crossing them with Rag1 mutant mice. The undertaken rotarod test revealed that the depletion of lymphocytes can enhance the cerebellar ataxic phenotype in female npc1–Rag1 double-knockout mice at all timepoints analyzed (4–7-wk-old mice; Fig 3A and B). The lack of lymphocytes did not affect the survival of the mutant mice (Fig S2A and B). We then counted the number of Purkinje cells in the cerebellum of 5-wk-old mutant mice, and we found a significant decrease of Purkinje cells in npc1–Rag1 double-knockout female mice when compared with those of npc1 single mutants (Fig 3C–G and C'–F'). Accordingly, we speculated that the

**Figure 1. Characterization of neurodegenerative phenotype and lifespan of *Npc1^nih* mice.**

**(A)** The accelerating rotarod test (accelerated from 0–50 rpm in 300 s) was used, and the latency to fall (in sec) was measured for npc1–/– mice and their wild-type npc1+/+ littermates in 27–69-d-old mice. Data were collected from both genders and are expressed as mean ± SEM (n = 20–25 mice per group). The legends for each group are indicated below. For the undertaking of statistical analysis, the two-tailed *t* test was used; *, *P* < 0.05; ***, *P* < 0.001. **(B, C)** Body weight was measured for female (B) and male (C) npc1–/– mice and their WT npc1+/+ littermates at 27–83 d of age. Data are expressed as mean ± SEM (n = 9–21 mice per group). The legends for each group are indicated below. For the

undertaking of statistical analysis, the two-tailed *t* test was used; *, *P* < 0.05; ***, *P* < 0.001. **(D, E, F, G, H, I)** Representative pictures of the immunostaining for calbindin+ Purkinje cells in the cerebella of 5- (panels (D, G)), 7- (panels (E, H)) and 10-wk-old (panels (F, I)) npc1+/+ (panels (D, E, F)) and npc1–/– mice (panels (G, H, I)). Scale bar in (D): 1 mm (applicable to panels (D, E, F, G, H, I)). The lobule number is indicated in (F). **(J)** The number of Purkinje cells counted in every lobule of the cerebella. Data were collected from both genders and are expressed as mean cell number per millimeter (length of Purkinje cell layer) ± SEM (n = 5–6 mice per group). The legends for each group are indicated below. For the undertaking of statistical analysis, the two-tailed *t* test was used; *, *P* < 0.05; **, *P* < 0.01; ***, *P* < 0.001 (for comparisons between the npc1+/+_7 wk and the npc1–/–_7 wk groups). ###, *P* < 0.001 (for comparisons between the npc1+/+_10 wk and the npc1–/–_10 wk groups). **(K)** Survival analysis for npc1–/– mice of both genders (n = 21 in female and n = 20 in male mice) as a function of age. For the undertaking of statistical analysis, log-rank test was used. No significant differences were observed between the groups (*P* = 0.604). **(L, M, N, O, P, Q, R, S)** Flow cytometry analysis for immune cells in the peripheral blood. The percentage of CD4+ ((L, P); designated as CD4/CD45 [%]), CD4+/CD25+/Foxp3–GFP+ ((M, Q); designated as Treg/CD45 [%]), CD11b+ ((N, R); designated as CD11b/CD45 [%]) and CD11b+/Ly6C+ cells ((O, S); designated as Ly6C/CD11b/CD45 [%]) over CD45+ cells in 5- and 7-wk-old female (L, M, N, O) and male mice (P, Q, R, S) is presented. Data are expressed as mean ± SEM (n = 10–12 mice per group in npc1+/+ groups; and n = 9–12 mice per group in npc1–/– groups). The legends for each group are indicated in (L, P). For the undertaking of statistical analysis, the two-tailed *t* test was used; *, *P* < 0.05; **, *P* < 0.01; ***, *P* < 0.001 (for comparisons between the npc1+/+ and the npc1–/– groups).

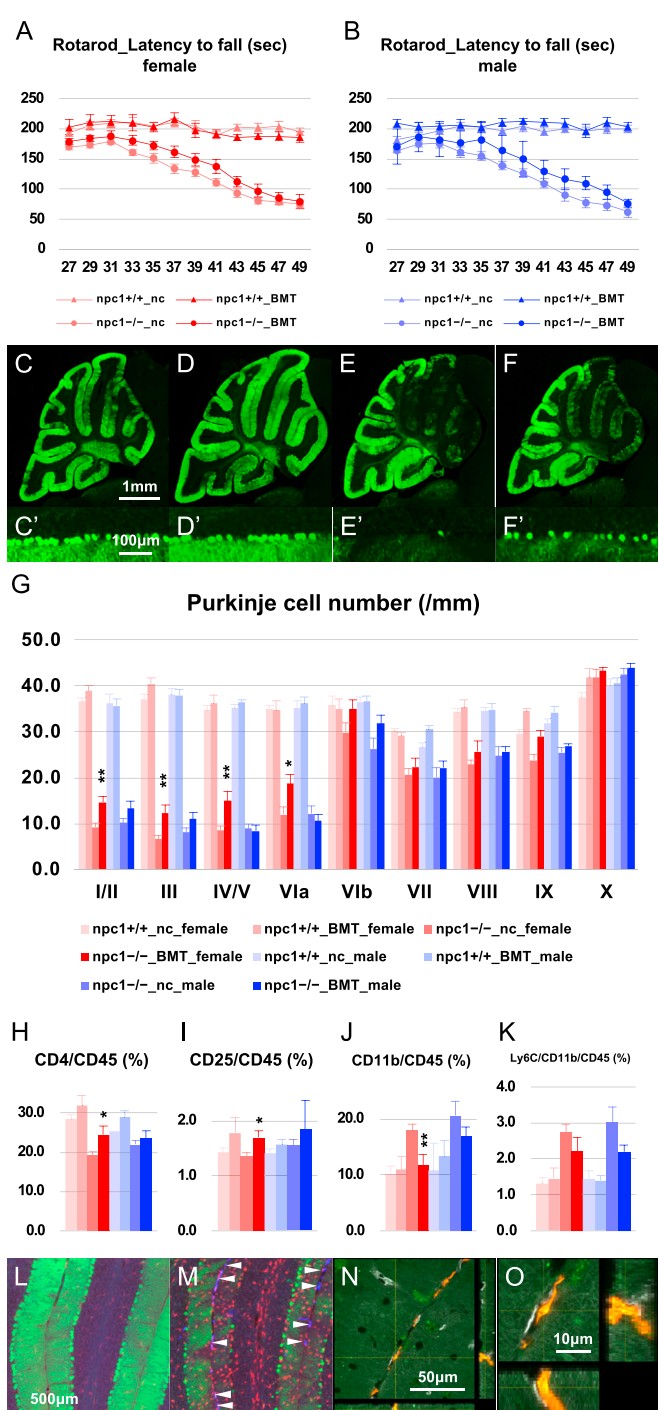

**Figure 2. Neuroprotective effect of neonatal bone marrow–derived cell transplantation.**

**(A, B)** The accelerating rotarod test (accelerated from 0–50 rpm in 300 s) was used, and the latency to fall (in sec) was measured for female (A) and male (B) npc1+/+_nc, npc1+/+_BMT, npc1−/−_nc, and npc1−/−_BMT groups in 27–49-d-old mice. Data are expressed as mean ± SEM (n = 5–10 mice per group in npc1+/+ groups; and n = 7–12 mice per group in npc1−/− groups). The legends for each group are indicated below. The mouse of which more than 10% of the CD11b-positive cells were positive for KuO were designated as "successfully BM-cell transplanted (BMT)" animal, and that of which less than 10% of the CD11b-posotive cells were positive for KuO were designated as "negative control (nc)." **(C, D, E, F, C', D', E', F')** Representative pictures of the immunostaining for calbindin⁺

decrease of the peripheral CD4-positive cells in npc1−/− mice (Fig 1L and P) can act as a trigger of neurodegeneration, and that the peripheral infusion of functional CD4-positive cells may prevent the Purkinje cell degeneration in NPC1 mutant mice.

## Peripheral infusion of CD4-positive cells prevented Purkinje cell death

The CD4- and CD8-positive T cells, used as a control, were isolated by fluorescence-activated cell sorting (FACS) from normal healthy 6–11-wk-old mice and were injected i.v. to NPC1 mutant mice (1–2 × 10⁵ cells/injection; twice a week for 3 wk; total of six injections in 4- to 7-wk-old mice). As a result, the female NPC1 mice that received CD4-positive T lymphocytes exhibited an improvement of cerebellar ataxia (Fig 4A and B). The timecourse and the extent of the ataxia in the CD8-positive cell–injected NPC1 mice had no difference from that of untreated NPC1 mice (Fig S2C). Subsequently, we performed the infusion experiment by injecting Th1 (CD4–CXCR3 double-positive), Th2 (CD4–CCR4 double-positive), Th17 (CD4–CCR6 double-positive), and Tregs (CD4–CD25 double-positive) in female mice (0.5–1 × 10⁴ cells/injection; twice a week for 3 wk; total of six injections in 4- to 7-wk-old mice). The undertaken rotarod test suggested that only the CD4–CD25 double-positive Tregs may have exerted a protective effect against neuronal degeneration (Fig S2C). We then isolated Tregs (CD4–CD25–GFP–triple-positive cells) from

---

Purkinje cells in the cerebella of 7-wk-old female npc1+/+_nc (C, C'), npc1+/+_BMT (D, D'), npc1−/−_nc (E, E'), and npc1−/−_BMT mice (F, F'). **(C, D, E, F, C', D', E', F')** Enlarged images of panels (C, D, E, F) are indicated in panels (C', D', E', F'), respectively. **(E, E', F, F')** Note that, Purkinje cells in anterior lobules are lost in npc1−/−_nc mice (panels (E, E')) and preserved in npc1−/−_BMT mice (panels (F, F')). Scale bars in (C): 1 mm (applicable to panels (C, D, E, F)) and in (C'): 100 μm (applicable to panels (C', D', E', F')). **(G)** The number of Purkinje cells counted in every lobule of the cerebella of 7-wk-old female and male mice. Data are expressed as mean cell number per millimeter (length of Purkinje cell layer) ± SEM (n = 5–7 mice per group in npc1+/+ groups; and n = 7–13 mice per group in npc1−/− groups). The legends for each group are indicated below. For the undertaking of statistical analysis, one-way ANOVA followed by Tukey HSD post hoc test was used for multiple comparison; *, P < 0.05; **, P < 0.01 (for comparisons between the npc1−/−_nc and the npc1−/−_BMT groups). **(H, I, J, K)** Flow cytometry analysis for immune cells in the peripheral blood of the mice. **(H, I, J, K)** The percentage of CD4⁺ ((H), designated as CD4/CD45 [%]), CD4⁺/CD25⁺ ((I), designated as CD25/CD45 [%]), CD11b⁺ ((J), designated as CD11b/CD45 [%]) and CD11b⁺/Ly6C⁺ cells ((K), designated as Ly6C/CD11b/CD45 [%]) over CD45⁺ cells in 7-wk-old female and male mice is presented. Data are expressed as mean ± SEM (n = 5–24 mice per group in npc1+/+ groups; and n = 7–13 mice per group in npc1−/− groups). **(G)** The legends for each group are indicated in (G). For the undertaking of statistical analysis, one-way ANOVA followed by Tukey HSD post hoc test was used for multiple comparison; *, P < 0.05; **, P < 0.01 (for comparisons between the npc1−/−_nc and the npc1−/−_BMT groups). **(L, M)** Immunohistochemical analysis for female npc1+/+_BMT (L) and npc1−/−_BMT mice (M). Sagittal sections of mouse cerebella were immunostained for Purkinje cells (anti-calbindin, green) and microglia/macrophages (anti-Iba1, red). **(M)** Note that the KuO⁺ bone marrow (BM)-derived cells (blue) were localized at the interlobular space of the cerebellum (indicated by arrowheads in panel (M)). Scale bar in (L): 500 μm (applicable to panels (L, M)). **(N, O)** Confocal images of the KuO⁺ BM-derived cells. Sagittal sections of female mouse cerebella were immunostained for monocytes/macrophages (anti-CD11b, green) and blood vessels (anti-Glut-1, gray). **(N, O)** An enlarged image of panel (N) is indicated in panel (O). Z-stacks were captured at 1.02-μm intervals by using a confocal laser–scanning microscope. Note that the KuO⁺ BM-derived cells (red) were CD11b⁺ macrophages and localized a perivascular zone of the interlobular space of the cerebellum. Scale bars in (N): 50 μm and (O): 10 μm.

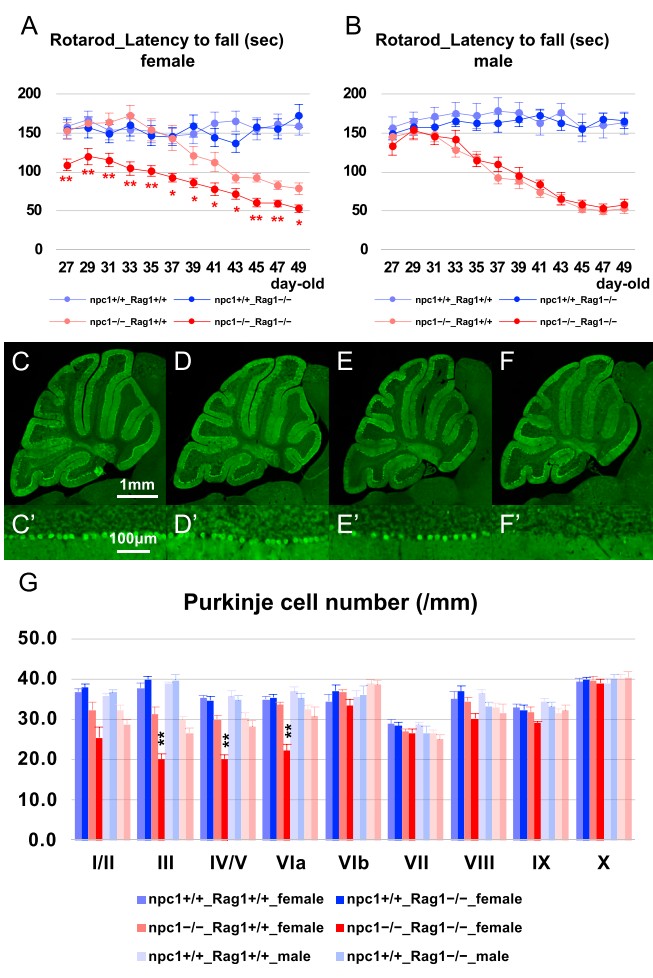

**Figure 3. Promotion of neurodegenerative phenotype by the lack of mature lymphocytes.**
**(A, B)** The accelerating rotarod test (accelerated from 0–50 rpm in 300 s) was used, and the latency to fall (in sec) was measured for female (A) and male (B) npc1+/+_Rag1+/+, npc1+/+_Rag1−/−, npc1−/−_Rag1+/+, and npc1−/−_Rag1−/− groups in 27–49-d-old mice. Data are expressed as mean ± SEM (n = 9–11 mice per group in npc1+/+ groups; and n = 10–12 mice per group in npc1−/− groups). The legends for each group are indicated below. For the undertaking of statistical analysis, the two-tailed t test was used; *, P < 0.05; **, P < 0.01 (for comparisons between the npc1-/-_Rag1+/+ and the npc1-/-_Rag1-/- groups). **(C, D, E, F, C', D', E', F')** Representative pictures of the immunostaining for calbindin+ Purkinje cells in the cerebella of 5-wk-old female npc1+/+_Rag1+/+ (C, C'), npc1+/+_Rag1−/− (D, D'), npc1−/−_Rag1+/+ (E, E'), and npc1−/−_Rag1−/− mice (F, F'). **(C, D, E, F, C', D', E', F')** Enlarged images of panels (C, D, E, F) are indicated in panels (C', D', E', F'), respectively. **(E, F, E', F')** Note that Purkinje cells in anterior lobules are lost in npc1−/−_Rag1−/− mice (panels (F, F')) compared with npc1−/−_Rag1+/+ mice (panels (E, E')). Scale bars in (C): 1 mm (applicable to panels (C, D, E, F)) and in (C'): 100 μm (applicable to panels (C', D', E', F')). **(G)** The number of Purkinje cells counted in every lobule of the cerebella of female and male mice. Data are expressed as mean cell number per millimeter (length of Purkinje cell layer) ± SEM (n = 4–6 mice per group in npc1+/+ groups; and n = 4–6 mice per group in npc1−/− groups). The legends for each group are indicated below. For the undertaking of statistical analysis, one-way ANOVA followed by Tukey HSD post hoc test was used for multiple comparison; **, P < 0.01 (for comparisons between the npc1−/−_ Rag1+/+ and the npc1−/−_ Rag1−/− groups).

the peripheral blood of healthy 6–11-wk-old Foxp3–GFP–Tg mice, and injected them (i.v.) to female NPC1 mice (0.5–1 × 10⁴ cells/injection; twice a week for 3 wk; total of six injections in 4- to 7-wk-

old mice). The improvement of cerebellar ataxia was similar to that observed in the whole CD4-positive cell-injected group (Fig 4A). The undertaken immunohistochemical analysis revealed a significant preservation of Purkinje cells by the treatment with whole CD4-positive cells and Foxp3-positive Tregs in female npc1−/− mice (Fig 4C–I and C'–H').

### Depletion of peripheral Treg cells enhanced Purkinje cell degeneration

To explore the possibility that a loss of functional Tregs can trigger neurodegeneration, we performed a rapid depletion of Tregs in NPC1 mice by using an anti-CD25 neutralizing antibody or Foxp3–diphtheria toxin receptor (DTR)–Tg mice. The injection of the anti-CD25 neutralizing antibody to npc1−/− mice (twice a week for 3 wk; total of six injections in 4- to 7-wk-old mice) resulted in a rapid aggravation of cerebellar ataxia in female 4–5-wk-old npc1−/− mice (Fig S2D and E); however, when the mice reached 5–7 wk of age, the timecourse and the extent of ataxia in the these mice was similar to that of the control IgG–injected female npc1−/− mice (Fig S2D). These findings suggest that Tregs can exert their neuroprotective effect during the early stages of neurodegeneration. Subsequently, we used Foxp3–DTR–Tg mice so as to specifically deplete Tregs, and we analyzed the occurring neurodegeneration in 4–5-wk-old mice. The rotarod test revealed an enhanced cerebellar ataxia in both female and male NPC1 mice (Fig 5A and B). The number of cerebellar Purkinje cells was significantly decreased in both female and male npc1−/−_Foxp3–DTR–Tg mice when compared with that of npc1−/−_non-Tg mice at 5 wk of age (Fig 5C–G and C'–F'). Not only the Foxp3-positive Tregs but also the whole CD4-positive cells were also decreased in the peripheral blood of both female and male npc1−/−_Foxp3–DTR–Tg mice (Fig 5H and I). We also found increased levels of CD11b-potitive and CD11b–Ly6C double-positive cells in female and male npc1−/−_Foxp3–DTR–Tg mice when compared with those of their npc1−/−_non-Tg control littermates (Fig 5J and K). Based on these findings, we hypothesized that CD11b-positive cells (specifically, CD11b–Ly6C double-positive inflammatory monocytes) may have infiltrated the CNS (Figs 2L–O and S1S–Z) and induced Purkinje cell degeneration.

### Depletion of the Ly6C-positive inflammatory monocytes prevented Purkinje cell degeneration

In an attempt to examine the involvement of peripheral Ly6C-positive inflammatory monocytes in the neurodegenerative process, we used Ccr2–red fluorescent protein (RFP)-knockin mice that harbor RFP cDNA to disrupt Ccr2 (designated as "Ccr2-RFP/RFP" or "Ccr2-R/R mice") so as to block the BM egress of Ly6C-positive monocytes. Analysis for the cerebellar ataxia through the rotarod test revealed an improvement in both female and male npc1−/−_Ccr2–RFP/RFP double-mutant mice when compared with npc1−/−_Ccr2+/+-single-mutant mice (Fig 6A and B). Immunohistochemical analysis revealed a significant preservation of Purkinje cells in both female and male npc1−/−_Ccr2–RFP/RFP double-mutant mice (Fig 6C–G and C'–F'). The npc1−/−_Ccr2–RFP/RFP double-mutant mice exhibited a decrease in the levels of their peripheral CD11b–Ly6C double-positive inflammatory monocytes (Fig 6K). Unexpectedly, peripheral CD4–CD25

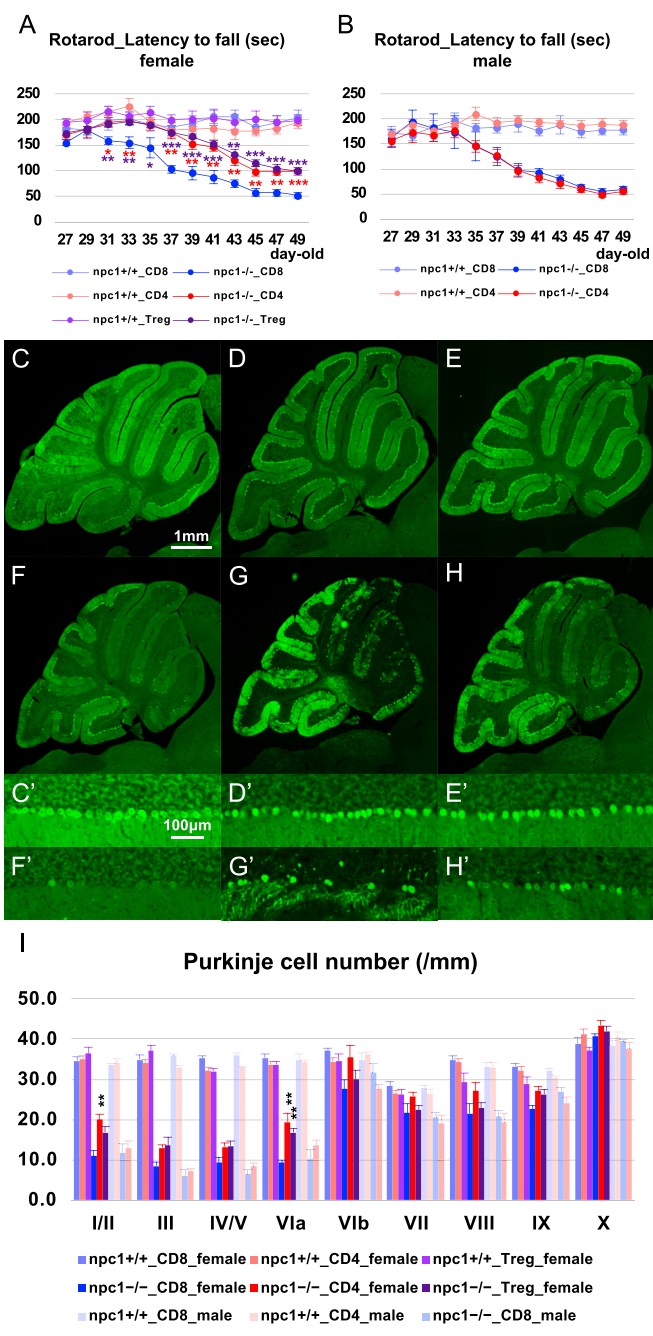

**Figure 4. Amelioration of neurodegenerative phenotype by infusion of CD4⁺ cells and Tregs.**

**(A, B)** The accelerating rotarod test (accelerated from 0–50 rpm in 300 s) was used, and the latency to fall (in sec) was measured for female npc1+/+_CD8, npc1+/+_CD4, npc1+/+_Treg, npc1−/−_CD8, npc1−/−_CD4, and npc1−/−_Treg groups (A) and male npc1+/+_CD8, npc1+/+_CD4, npc1−/−_CD8, and npc1−/−_CD4 groups (B) in 27–49-d-old mice. Data are expressed as mean ± SEM (n = 4–8 mice per group in npc1+/+ groups; and n = 5–9 mice per group in npc1−/− groups). The legends for each group are indicated below. For the undertaking of statistical analysis, the two-tailed t test was used; *, P < 0.05; **, P < 0.01; ***, P < 0.001 (for comparisons between npc1−/−_CD8 mice and other groups). **(C, D, E, F, G, H, C', D', E', F', G', H')** Representative pictures of the immunostaining for calbindin⁺ Purkinje cells in the cerebella of 7-wk-old female npc1+/+_CD8 (C, C'), npc1+/+_CD4 (D, D'), npc1+/+_Treg (E, E'), npc1−/−_CD8 (F, F'), npc1−/−_CD4 (G, G'), and npc1−/−_Treg (H, H'). **(C, D, E, F, G, H, C', D', E', F', G', H')** Enlarged images of panels

(C, D, E, F, G, H) are indicated in panels (C', D', E', F', G', H'), respectively. **(F, G, H, F', G', H')** Note that Purkinje cells in anterior lobules are preserved in npc1−/−_CD4 (panels (G, G')) and npc1−/−_Treg mice (panels (H, H')) compared with npc1−/−_CD8 mice (panels (F, F')). Scale bars in (C): 1 mm (applicable to panels (C, D, E, F, G, H)) and in (C'): 100 μm (applicable to panels (C', D', E', F', G', H')). **(I)** The number of Purkinje cells counted in every lobule of the cerebella of female and male mice. Data are expressed as mean cell number per millimeter (length of Purkinje cell layer) ± SEM (n = 4–7 mice per group in npc1+/+ groups; and n = 5–9 mice per group in npc1−/− groups). The legends for each group are indicated below. For the undertaking of statistical analysis, one-way ANOVA followed by Tukey HSD post hoc test was used for multiple comparison; **, P < 0.01 (for comparisons between npc1−/−_CD8 mice and other groups).

double-positive cells were not found decreased in either female or male npc1−/−_Ccr2−RFP/RFP double-mutant mice (Fig 6I). The levels of the peripheral CD4-positive cells did not decrease, whereas those of the CD11b-positive cells did not increase in female npc1−/−_Ccr2−RFP/RFP double-mutant mice (Fig 6H and J). Finally, the loss of Ly6C-potitive inflammatory monocytes did not affect the survival of mutant mice (Fig S2F and G).

We subsequently asked whether peripheral monocytes display a distinct disease-associated gene expression signature. To this end, RNA-seq was performed in peripheral CD11b-positive monocytes isolated from 7-wk-old female npc1−/− and from npc1+/+ control littermates (n = 3 each; Fig S3A). The correlation and principal component analysis revealed a transcriptional divergence between the two groups (Fig S3B and C). Differentially expressed genes were identified at a false discovery rate (FDR) of 0.05. A total of 512 genes were found to be up-regulated and 148 genes were found to be down-regulated in the npc1−/− monocytes when compared with npc1+/+ monocytes (Fig S3D). The top 30 up-regulated genes are highlighted in Fig S3E and F. We confirmed that cholesterol syn-thesis genes such as *Srebf2* (FDR, 0.00053033), *Hmgcr* (0.01094964), and *Hmgcs1* (0.00101991) were overexpressed in NPC1 monocytes. Gene Ontology (GO) enrichment analysis for the up-regulated genes included the immune system process (adjusted *P*-value: $1.06 \times 10^{-27}$; Term ID: GO:0002376), cell migration ($5.91 \times 10^{-23}$; GO:0016477), the inflammatory response ($4.57 \times 10^{-14}$; GO:0006954), programmed cell death ($1.01 \times 10^{-12}$; GO:0012501), the lipid metabolic process ($4.49 \times 10^{-12}$; GO:0006629), leukocyte activation ($1.35 \times 10^{-11}$; GO:0045321), and the regulation of cytokine production ($2.54 \times 10^{-11}$; GO:0001817) (Table S1). The DEGs in the selected GO terms are listed in Table S1. On the other hand, the GO enrichment analysis for the down-regulated genes was linked to biological processes involved in the interspecies interaction between organisms ($6.50 \times 10^{-3}$; GO:0044419) and to the response to biotic stimulus ($3.74 \times 10^{-2}$; GO:0009607). We further performed GO analysis using a different tool ShinyGO (Ge et al, 2020). We found several critical biological pro-cesses including steroid biosynthesis (enrichment FDR: $2.43 \times 10^{-4}$), cholesterol metabolism ($1.56 \times 10^{-2}$), and cytokine–cytokine re-ceptor interaction ($6.30 \times 10^{-3}$) were enriched (Table S2).

We investigated the age-dependent changes in the transcrip-tional expression of several key molecules that could be involved in the NPC1 pathology, through the undertaking of quantitative RT–PCR on whole cerebellar tissues (Fig S4). The data are expressed as mean values relative to the housekeeping gene (*Gapdh*) that has been used as a reference gene in previous works in NPC1 mice (Cluzeau et al, 2012; Cologna et al, 2014), and they are normalized to

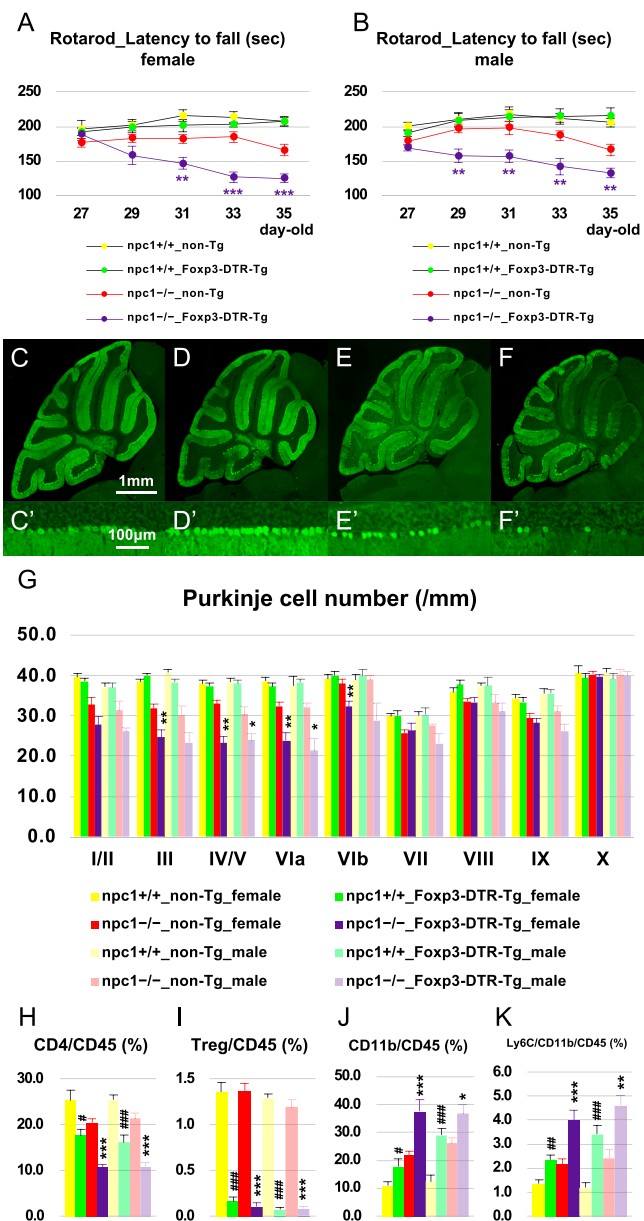

**Figure 5. Promotion of neurodegenerative phenotype by the lack of Tregs.**
**(A, B)** The accelerating rotarod test (accelerated from 0 to 50 rpm in 300 s) was used, and the latency to fall (in sec) was measured for female (A) and male (B) npc1+/+_non-Tg, npc1+/+_Foxp3–DTR–Tg, npc1–/–_non-Tg, and npc1–/–_Foxp3–DTR–Tg groups in 27–35-d-old mice. Data are expressed as mean ± SEM (n = 8–11 mice per group in npc1+/+ groups; and n = 10–13 mice per group in npc1–/– groups). The legends for each group are indicated below. For the undertaking of statistical analysis, the two-tailed t test was used; **, P < 0.01; ***, P < 0.001 (for comparisons between the npc1–/–_non-Tg and the npc1–/–_Foxp3–DTR–Tg groups). **(C, D, E, F, C', D', E', F')** Representative pictures of the immunostaining for calbindin⁺ Purkinje cells in the cerebella of 5-wk-old female npc1+/+_non-Tg (C, C'), npc1+/+_Foxp3–DTR–Tg (D, D'), npc1–/–_non-Tg (E, E') and npc1–/–_Foxp3–DTR–Tg (F, F'). **(C, D, E, F, C', D', E', F')** Enlarged images of panels (C, D, E, F) are indicated in panels (C', D', E', F'), respectively. **(E, F, E', F')** Note that Purkinje cells in anterior lobules are lost in npc1–/–_Foxp3–DTR–Tg (panels (F, F')) compared with npc1–/–_non-Tg mice (panels (E, E')). Scale bars in (C): 1 mm (applicable to panels (C, D, E, F)) and in (C'): 100 μm (applicable to panels (C', D', E', F')). **(G)** The number of Purkinje cells counted in every lobule of the cerebella of female and male mice. Data are expressed as mean cell number per millimeter (length of Purkinje cell layer) ± SEM (n = 5 mice per group in

the npc1+/+_5 wk group. A decrease of the transcripts for *Calbindin-1* and an increase of the transcripts for *Aif1* (encoding the ionized calcium–binding adapter molecule 1 or Iba1; a microglia/macrophage marker) and *Cx3cr1* were observed (Fig S4A–C). *ApoE* and *Lgals3* (encoding galectin-3) were also found to be significantly up-regulated in the npc1–/– cerebellum (Fig S4G and H) as previously reported (Li et al, 2005; Cluzeau et al, 2012). Several genes encoding chemokines (*Ccl2*, *Ccl5*, and *Cxcl10*) (Fig S4D–F), matrix metalloproteinases (*Mmp2*, *Mmp3*, *Mmp9*, and *Mmp12*) (Fig S4I–L), tissue inhibitors of metalloproteinases (*Timp1*, *Timp2*, and *Timp3*) (Fig S4M–O), and cell adhesion molecules (*Icam1*, *Vcam1*, and *Alcam*) (Fig S4P–R) were found to be up-regulated.

## Depletion of brain-resident microglia did not affect neurodegeneration

Finally, we addressed an involvement of the resident innate immune cells, the microglial cells, in the neurodegenerative process of NPC1. The immunoreactivity for Tmem119, a homeostatic microglia-specific protein (Bennett et al, 2016), was found reduced in most of the Iba1-positive cerebellar microglia in npc1–/– mice (Fig S5A–F), thereby suggesting the occurrence of a switching to a disease-associated microglia (DAM)-like phenotype (Keren-Shaul et al, 2017). We used two independent strategies to deplete microglia: the ablation of *Csf-1* and the oral administration of the CSF-1 receptor (CSF-1R) inhibitor PLX3397.

We first analyzed the occurring neurodegenerative changes by using osteopetrotic (op/op) mutant mice that are characterized by a defective CSF-1 production (designated as "Csf-1op/op mice"). These mice exhibit a depletion of microglial cells, that is, relatively specific to the cerebellum and the brainstem but not the forebrain region where the alternate CSF-1R ligand IL-34 is distributed (Greter et al, 2012; Nandi et al, 2012; Wang et al, 2012; Nakamichi et al, 2013; Mammana et al, 2018; Kana et al, 2019). As we were not able to obtain npc1–/–_Csf-1op/op-double mutant mice, we produced npc1–/–_Csf-1op/+ mutant mice that have a hemizygous *Csf-1* mutation. The npc1–/–_Csf-1op/+ mice displayed fewer microglia in the cerebellum (Fig 7A–F). The cerebellar ataxia (as analyzed by a rotarod test) did not show any significant differences in either female or male npc1–/–_Csf-1op/+ mutant mice when compared with that of npc1–/–_Csf-1+/+ control mice (Fig 8A and B). The

npc1+/+ groups; and n = 5–11 mice per group in npc1–/– groups). The legends for each group are indicated below. For the undertaking of statistical analysis, one-way ANOVA followed by Tukey HSD post hoc test was used for multiple comparison; *, P < 0.05; **, P < 0.01 (for comparisons between the npc1–/–_non-Tg and the npc1–/–_Foxp3–DTR–Tg groups). **(H, I, J, K)** Flow cytometry analysis for immune cells in the peripheral blood of the mice. **(H, I, J, K)** The percentage of CD4⁺ ((H), designated as CD4/CD45 [%]), CD4⁺/CD25⁺ ((I), designated as CD25/CD45 [%]), CD11b⁺ ((J), designated as CD11b/CD45 [%]) and CD11b⁺/Ly6C⁺ cells ((K), designated as Ly6C/CD11b/CD45 [%]) over CD45⁺ cells in 7-wk-old female and male mice is presented. Data are expressed as mean ± SEM (n = 7–16 mice per group in npc1+/+ groups; and n = 6–16 mice per group in npc1–/– groups). **(G)** The legends for each group are indicated in (G). For the undertaking of statistical analysis, one-way ANOVA followed by Tukey HSD post hoc test was used for multiple comparison; #, P < 0.05; ##, P < 0.01; ###, P < 0.001 (for comparisons between the npc1+/+_non-Tg and the npc1+/+_Foxp3–DTR–Tg groups) and *, P < 0.05; **, P < 0.01; ***, P < 0.001 (for comparisons between the npc1–/–_non-Tg and the npc1–/–_Foxp3–DTR–Tg groups).

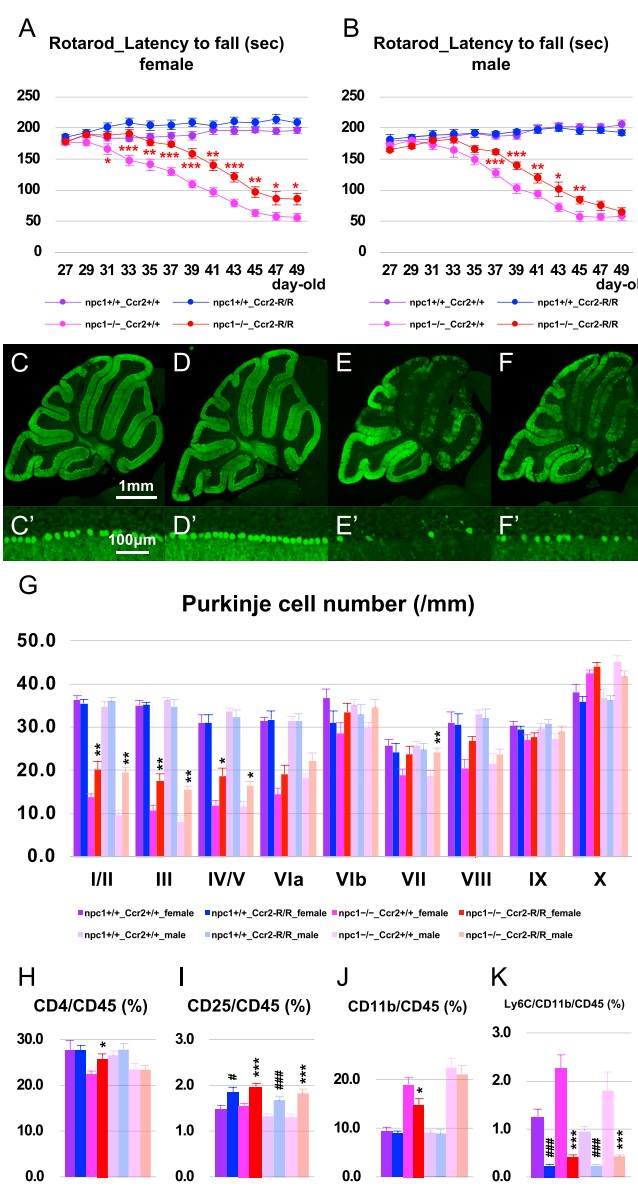

**Figure 6. Amelioration of neurodegenerative phenotype by the lack of inflammatory monocytes.**
**(A, B)** The accelerating rotarod test (accelerated from 0–50 rpm in 300 s) was used, and the latency to fall (in sec) was measured for female (A) and male (B) npc1+/+_Ccr2+/+, npc1+/+_Ccr2-R/R, npc1−/−_Ccr2+/+, and npc1−/−_Ccr2-R/R groups in 27–49-d-old mice. Data are expressed as mean ± SEM (n = 10 mice per group in npc1+/+ groups; and n = 10–14 mice per group in npc1−/− groups). The legends for each group are indicated below. For the undertaking of statistical analysis, the two-tailed *t* test was used; *, *P* < 0.05; **, *P* < 0.01; ***, *P* < 0.001 (for comparisons between the npc1−/−_Ccr2+/+ and the npc1−/−_Ccr2-R/R groups). **(C, D, E, F, C', D', E', F')** Representative pictures of the immunostaining for calbindin+ Purkinje cells in the cerebella of 7-wk-old npc1+/+_Ccr2+/+ (C, C'), npc1+/+_Ccr2-R/R (D, D'), npc1−/−_Ccr2+/+ (E, E'), and npc1−/−_Ccr2-R/R (F, F'). **(C, D, E, F, C', D', E', F')** Enlarged images of panels (C, D, E, F) are indicated in panels (C', D', E', F'), respectively. **(E, F, E', F')** Note that Purkinje cells in anterior lobules are preserved in npc1−/−_Ccr2-R/R (panels (F, F')) compared with npc1−/−_Ccr2+/+ mice (panels (E, E')). Scale bars in (C): 1 mm (applicable to panels (C, D, E, F)) and in (C'): 100 μm (applicable to panels (C', D', E', F')). **(G)** The number of Purkinje cells counted in every lobule of the cerebella of female and male mice. Data are expressed as mean cell number per millimeter (length of Purkinje cell layer) ± SEM (n = 6 mice per group in npc1+/+ groups; and n = 10–14 mice per group in npc1−/− groups). The legends for each group are indicated below. For the

numbers of cerebellar Purkinje cells were also unaffected by the depletion of the microglia in these mutant mice (Fig 8C–G and C'–F'). The peripheral immune cells (including CD4-positve, CD4–CD25 double-positive, and CD11b-positive cells) were unaffected by the decrease of the CSF1 production (Fig 8H–J). Although the levels of the CD11b–Ly6C double-positive cells were found decreased in the npc1+/+_Csf-1op/+ mutant mice (when compared with those of npc1+/+_Csf-1+/+ ones), the difference was non-significant (Fig 8K).

Subsequently, we used the oral CSF-1R inhibitor PLX3397 to deplete the microglia in the whole brain (Elmore et al, 2014). Mice (4–7 or 6–7-wk-old) were administered with PLX3397 to study the effect of age and the duration of the microglial loss. The treatment induced a decrease in the number microglia in the cerebellum in both regimens (Fig 9A–H). The cerebellar ataxia was slightly improved in female npc1−/−_PLX3397_4-7 and npc1−/−_PLX3397_6-7 mice and in male npc1−/−_PLX3397_4-7 mice when compared with that of npc1−/−_normal diet control mice (Fig 10A and B). The number of cerebellar Purkinje cells was not affected significantly by the depletion of microglia in these mice (Fig 10C–I and C'–H'). The levels of the peripheral immune cells (including CD4-positve, CD4–CD25 double-positive, CD11b-positive, and CD11b–Ly6C double-positive cells) were undisturbed by the inhibition of the CSF-1R (Fig S5G–J). These findings imply that the microglial cells have little effect on the observed Purkinje cell degeneration in NPC1 mice.

## Discussion

In this study, we investigated whether the peripheral adaptive and the innate immune systems are involved in the neurodegenerative process of NPC1. We aimed at providing a mechanism-based therapeutic strategy that can potentially be combined with on-going lipid reduction therapies (i.e., miglustat and HPβCD). We discovered a neuroprotective effect of neonatal BM–derived cell transplantation, and this finding led us to determine which cell populations are involved in the promotion of Purkinje cell survival. A previous study has found no change in motor activity, weight loss, or survival in BM-derived cell transplantation at 3–4-wk-old (Cougnoux et al, 2018a). Based on the finding that the brain-resident microglia were activated at 7-d-old (Colombo et al, 2021), we speculated that the age of recipients and/or lethal irradiation in their transplantation experiment might have affected

undertaking of statistical analysis, one-way ANOVA followed by Tukey HSD post hoc test was used for multiple comparison; *, *P* < 0.05; **, *P* < 0.01 (for comparisons between the npc1−/−_Ccr2+/+ and the npc1−/−_Ccr2-R/R groups). **(H, I, J, K)** Flow cytometry analysis for immune cells in the peripheral blood of the mice. The percentage of CD4+ ((H), designated as CD4/CD45 [%]), CD4+/CD25+ ((I), designated as CD25/CD45 [%]), CD11b+ ((J), designated as CD11b/CD45 [%]) and CD11b+/Ly6C+ cells ((K), designated as Ly6C/CD11b/CD45 [%]) over CD45+ cells in 7-wk-old female and male mice is presented. Data are expressed as mean ± SEM (n = 10–14 mice per group in npc1+/+ groups; and n = 10–14 mice per group in npc1−/− groups). **(G)** The legends for each group are indicated in (G). For the undertaking of statistical analysis, one-way ANOVA followed by Tukey HSD post hoc test was used for multiple comparison; #, *P* < 0.05; ###, *P* < 0.001 (for comparisons between the npc1+/+_Ccr2+/+ and the npc1+/+_Ccr2-R/R groups) and *, *P* < 0.05; ***, *P* < 0.001 (for comparisons between the npc1−/−_Ccr2+/+ and the npc1−/−_Ccr2-R/R groups).

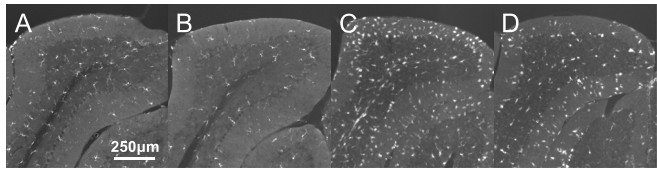

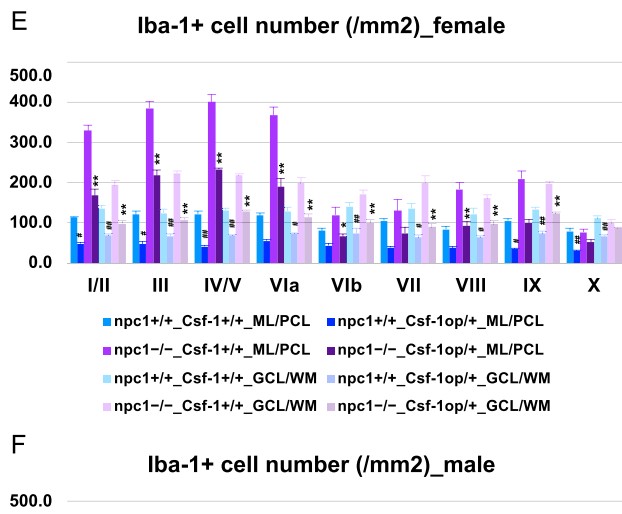

**E**

**Iba-1+ cell number (/mm2)_female**

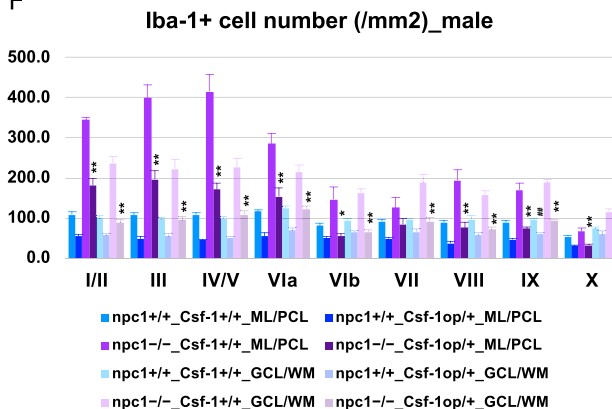

- npc1+/+_Csf-1+/+_ML/PCL
- npc1+/+_Csf-1op/+_ML/PCL
- npc1−/−_Csf-1+/+_ML/PCL
- npc1−/−_Csf-1op/+_ML/PCL
- npc1+/+_Csf-1+/+_GCL/WM
- npc1+/+_Csf-1op/+_GCL/WM
- npc1−/−_Csf-1+/+_GCL/WM
- npc1−/−_Csf-1op/+_GCL/WM

**F**

**Iba-1+ cell number (/mm2)_male**

- npc1+/+_Csf-1+/+_ML/PCL
- npc1+/+_Csf-1op/+_ML/PCL
- npc1−/−_Csf-1+/+_ML/PCL
- npc1−/−_Csf-1op/+_ML/PCL
- npc1+/+_Csf-1+/+_GCL/WM
- npc1+/+_Csf-1op/+_GCL/WM
- npc1−/−_Csf-1+/+_GCL/WM
- npc1−/−_Csf-1op/+_GCL/WM

**Figure 7. A partial depletion of microglia by hemizygous mutation of *Csf-1* in *Npc1^nih* mice.**

**(A, B, C, D)** Representative pictures of the immunostaining for Iba1+ microglia/macrophages in the cerebella (lobules III and IV/V) of 7-wk-old female npc1+/+ _Csf-1+/+ (A), npc1+/+_Csf-1op/+ (B), npc1−/−_Csf-1+/+ (C), and npc1−/−_Csf-1op/+ (D). **(A, B, C, D)** Note that Iba1+ cells are decreased in npc1+/+_Csf-1op/+ (B) and npc1−/−_Csf-1op/+ (D) compared with npc1+/+_Csf-1+/+ (A) and npc1−/−_Csf-1+/+ mice (C), respectively. Scale bars in (A): 250 μm (applicable to panels (A, B, C, D)). **(E, F)** The number of Iba1+ cells counted in molecular layer (ML) and Purkinje cell layer (PCL; designated as ML/PCL) or granule cell layer (GCL) and white matter (WM; designated as GCL/WM) in every lobule of the cerebella of female (E) and male mice (F). Data are expressed as mean cell number per square millimeter (area of ML/PCL or GCL/WM) ± SEM (n = 4 mice per group in npc1+/+ groups; and n = 6 mice per group in npc1−/− groups). The legends for each group are indicated below. For the undertaking of statistical analysis, one-way ANOVA followed by Tukey HSD post hoc test was used for multiple comparison; #, $P < 0.05$; ##, $P < 0.01$ (for comparisons between the npc1+/+_Csf-1+/+ and the npc1+/+_Csf-1op/+ groups) and *, $P < 0.05$; **, $P < 0.01$ (for comparisons between the npc1−/−_Csf-1+/+ and the npc1−/−_Csf-1op/+ groups).

the inflammatory state of peripheral immune cells. Because the decrease of peripheral CD4-positive cells was found attenuated in the BM cell–transplanted female mice (Fig 2H), and conversely, because the depletion of mature T and B lymphocytes (Fig 3) or Tregs (Fig 5) was found to enhance neuronal degeneration, we

examined the effect of the infusion of CD4-poitive cells or Tregs from healthy donors to NPC1 mice (Fig 4). Our experimental findings suggested that a malfunction of CD4-positive cells and/or Tregs during an early stage of the disease may act as a trigger for the Purkinje cell degeneration observed in NPC1. In a mouse model of ALS, a fatal neuromuscular disease characterized by motoneuron degeneration in the brain and the spinal cord, the lack of functional lymphocytes is known to accelerate the disease progression (Beers et al, 2008; Chiu et al, 2008). The recruitment of reconstituted lymphocytes from BM transplantation to the sites of injury provides neuroprotection by modulating the beneficial function of microglia (Beers et al, 2008; Chiu et al, 2008). Other studies have revealed that the passive transfer of CD4-positive T cells or Tregs with or without an ex vivo activation can extend the survival of ALS mice (Banerjee et al, 2008; Beers et al, 2011). Furthermore, in ALS patients, the decreased levels of blood CD4-positive T lymphocytes have been associated with cognitive impairment (Yang et al, 2021), whereas the Tregs were found to be dysfunctional, with impaired immunosuppressive ability (Henkel et al, 2013; Beers et al, 2017). Although a recent small clinical trial of aHSCT did not reveal a significant modification of the disease progression (Lunetta et al, 2022), the infusion of expanded autologous Tregs has slowed the progression of the disease and has stabilized the inspiratory pressure in ALS patients (Thonhoff et al, 2018; Rajabinejad et al, 2020). The passive transfer of Tregs that were activated ex vivo was also able to suppress the toxic microglial responses, to up-regulate neurotrophic factors, and to protect dopaminergic neurons in PD model mice (Reynolds et al, 2007, 2010). Based on these findings, the delivery of CD4-positive T lymphocytes and/or Tregs may have a potential to treat NPC1, although further studies are required to explore the precise target(s) and site(s) of action of these cells in NPC1.

Gender-specific differences in the cerebellar Purkinje cell degeneration have been previously reported in NPC1 (Võikar et al, 2002; Holzmann et al, 2021), heterozygous staggerer (Lemaigre-dubreuil et al, 1999), and reeler mice (Hadj-Sahraoui et al, 1996). In these studies, the Purkinje cell loss and the related behavioral abnormalities were more severe in male than female mice; facts are consistent with our current observation. Previous reports showed the effects of sex hormones on adaptive immune cells (Klein & Flanagan, 2016; Dodd & Menon, 2022). Deprivation of androgen and knockout of androgen receptors in mice resulted in thymic enlargement, indicating that androgens have a suppressive effect on lymphopoiesis in early development (Roden et al, 2004; Lai et al, 2012; Dodd & Menon, 2022). Females have higher numbers of total and activated CD4-positive T cells than males (Abdullah et al, 2012; Sankaran-Walters et al, 2013; Klein & Flanagan, 2016). Moreover, estrogen and progesterone have ability to augment expression of Foxp3 in CD4-positive cells in mice and humans (Polanczyk et al, 2004; Lee et al, 2011). These immune-modulating effects of sex hormones may potentially be responsible for the gender-specific neuronal protection and degeneration observed in this study. The NPC1 patients display no gender-based differences, but they do display an age-dependent heterogeneity of onset, progression, and symptoms of the disease (Vanier, 2010; Stampfer et al, 2013; Wraith et al, 2014; Mengel et al, 2017). Moreover, no gender-dependent survival differences have been found in NPC1 patients (Bianconi

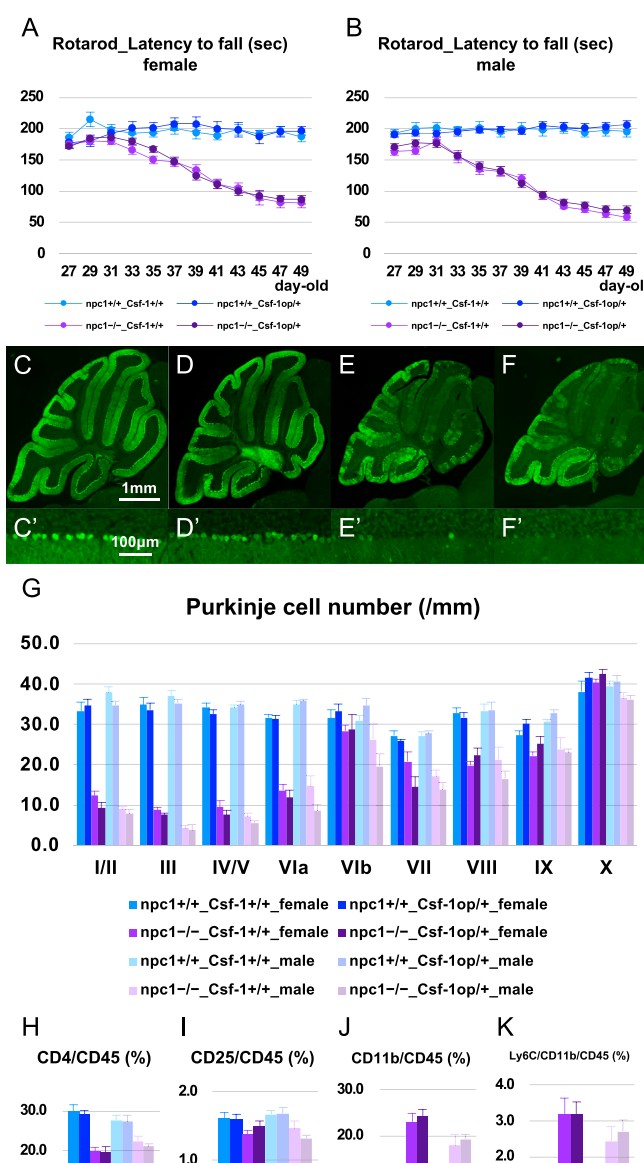

**Figure 8. Non-significant effect on neurodegenerative phenotype with a partial depletion of microglia by hemizygous mutation of *Csf-1*.**

**(A, B)** The accelerating rotarod test (accelerated from 0–50 rpm in 300 s) was used, and the latency to fall (in seconds) was measured for female (A) and male (B) npc1+/+_Csf-1+/+, npc1+/+_Csf-1op/+, npc1–/–_Csf-1+/+, and npc1–/–_Csf-1op/+ groups in 27–49-d-old mice. Data are expressed as mean ± SEM (n = 8–12 mice per group in npc1+/+ groups; and n = 10–15 mice per group in npc1–/– groups). The legends for each group are indicated below. For the undertaking of statistical analysis, the two-tailed *t* test was used. No significant differences were identified between npc1–/–_Csf-1+/+ and npc1–/–_Csf-1op/+ groups in either gender. **(C, D, E, F, C', D', E', F')** Representative pictures of the immunostaining for calbindin⁺ Purkinje cells in the cerebella of 7-wk-old female npc1+/+_Csf-1+/+ (C, C'), npc1+/+_Csf-1op/+ (D, D'), npc1–/–_Csf-1+/+ (E, E'), and npc1–/–_Csf-1op/+ (F, F'). **(C, D, E, F, C', D', E', F')** Enlarged images of panels (C, D, E, F) are indicated in panels (C', D', E', F'), respectively. **(E, F, E', F')** Note that Purkinje cells in anterior lobules are lost to a similar extent in npc1–/–_Csf-1+/+ (E, E') and npc1–/–_Csf-1op/+ mice (F, F'). Scale bars in (C): 1 mm (applicable to panels (C, D, E, F)) and in (C'): 100 μm (applicable to panels (C', D', E', F')). **(G)** The number of Purkinje cells counted in every lobule of the cerebella of female and male mice. Data are expressed as mean cell number per millimeter (length of Purkinje cell layer) ±

et al, 2019). We believe that the therapeutic effects observed herein in female mice represent a promising basis for the development of new treatments for NPC1 patients.

Previous studies have reported a functional impairment of peripheral adaptive immune cells in NPC1 (Platt et al, 2016). Speak et al (2014) have hypothesized that the lysosomal accumulation of sphingosine may compromise the systemic gradient of chemotactic sphingosine-1-phosphate (Speak et al, 2014), which is an important factor for the tissue trafficking of lymphocytes (Cyster & Schwab, 2012; MacEyka & Spiegel, 2014). They have shown an altered tissue distribution of NK cells in NPC1 mice, where these cells were retained in multiple lymphoid tissues and their circulating levels were found reduced (Speak et al, 2014). Newton et al (2017) demonstrated that FTY720/fingolimod, an inhibitor of histone deacetylase, enhanced expression of NPC1 and decreased accumulation of cellular cholesterol in NPC1 mutant human fibroblasts (Newton et al, 2017). This report provided a potential therapeutic opportunity to treat the patients with NPC. However, FTY720/fingolimod acts as a modulator of sphingosine-1-phosphate receptor to prevent the egress of lymphocytes from secondary lymphoid organs and is frequently used for the treatment of MS (McGinley & Cohen, 2021; Dumitrescu et al, 2023). The neuroprotective property of FTY720/fingolimod should be carefully evaluated in NPC1 mutant animals. Moreover, because of a reduced content/release of lysosomal calcium in NPC1, the NK cells are defective in facilitating the degranulation of cytotoxic granules, thereby leading to reduced cytotoxicity (Speak et al, 2014). NK cells from NPC patients exhibited a similar alteration of frequency and phenotype (Speak et al, 2014). Moreover, a recent study has shown that the NPC1 deficiency impairs the pore formation of perforin and the target-killing ability of cytotoxic T lymphocytes (Castiblanco et al, 2022). The impairment of these cytotoxic cells may cause atypical infection and pulmonary complications, often contributing to the cause of death of NPC patients (Imrie et al, 2007). Lysosomal calcium defects can also result to an impaired platelet aggregation and a prolonged bleeding time in NPC1 mice (Chen et al, 2020). Our study has indicated that the adaptive immune system is more broadly affected and can be a target of future clinical interventions in NPC1.

We have also found that the Purkinje cell death was ameliorated by depleting the circulating inflammatory monocytes (Fig 6). This finding suggests that macrophages infiltrating into the perivascular zones of the interlobular space (Figs 2L–O and S1S–Z), the ML, and

SEM (n = 6 mice per group in npc1+/+ groups; and n = 6 mice per group in npc1–/– groups). The legends for each group are indicated below. For the undertaking of statistical analysis, one-way ANOVA followed by Tukey HSD post hoc test was used for multiple comparison. No significant differences were identified between npc1–/–_Csf-1+/+ and npc1–/–_Csf-1op/+ groups in either gender. **(H, I, J, K)** Flow cytometry analysis for immune cells in the peripheral blood of the mice. The percentage of CD4⁺ ((H), designated as CD4/CD45 [%]), CD4⁺/CD25⁺ ((I), designated as CD25/CD45 [%]), CD11b⁺ ((J), designated as CD11b/CD45 [%]) and CD11b⁺/Ly6C⁺ cells ((K), designated as Ly6C/CD11b/CD45 [%]) over CD45⁺ cells in 7-wk-old female and male mice is presented. Data are expressed as mean ± SEM (n = 11–20 mice per group in npc1+/+ groups; and n = 11–17 mice per group in npc1–/– groups). **(G)** The legends for each group are indicated in (G). For the undertaking of statistical analysis, one-way ANOVA followed by Tukey HSD post hoc test was used for multiple comparison; *, *P* < 0.05; **, *P* < 0.01 (for comparisons between the npc1+/+_Csf-1+/+ and the npc1+/+_Csf-1op/+ groups).

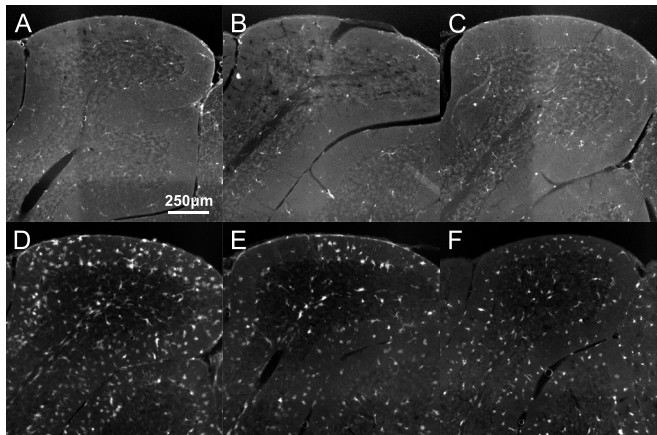

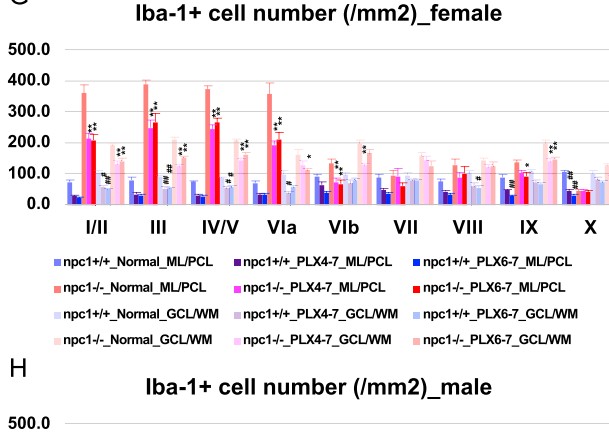

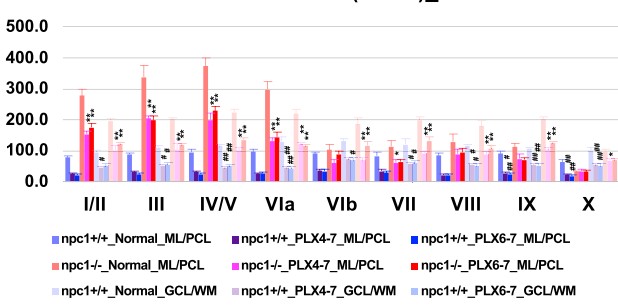

**Figure 9. A partial depletion of microglia by oral administration of CSF-1R inhibitor PLX3397 in *Npc1^nih* mice.**

**(A, B, C, D, E, F)** Representative pictures of the immunostaining for Iba1⁺ microglia/macrophages in the cerebella (lobules III and IV/V) of 7-wk-old female npc1+/+_Normal ((A), normal diet), npc1+/+_PLX4-7 ((B), PLX3397 treatment at 4–7 wk of age), npc1+/+_PLX6-7 ((C), PLX3397 treatment at 6–7 wk of age), npc1–/–_Normal (D), npc1–/–_PLX4-7 (E) and npc1–/–_PLX6-7 (F). **(A, B, C, D, E, F)** Note that Iba1⁺ cells are decreased in npc1+/+_PLX4-7 (B) and npc1+/+_PLX6-7 (C) compared with npc1+/+_Normal (A), and npc1–/–_PLX4-7 (E) and npc1–/–_PLX6-7 (F) compared with npc1–/–_Normal (D). Scale bars in (A): 250 μm (applicable to panels (A, B, C, D, E, F)). **(G, H)** The number of Iba1⁺ cells counted in molecular layer (ML) and Purkinje cell layer (PCL; designated as ML/PCL) or granule cell layer (GCL) and white matter (WM; designated as GCL/WM) in every lobule of the cerebella of female (G) and male mice (H). Data are expressed as mean cell number per square millimeter (area of ML/PCL or GCL/WM) ± SEM (n = 4 mice per group in npc1+/+ groups; and n = 6 mice per group in npc1–/– groups). The legends for each group are indicated below. For the undertaking of statistical analysis, one-way ANOVA followed by Tukey HSD post hoc test was used for multiple comparison; #, $P < 0.05$; ##, $P < 0.01$ (for comparisons between npc1+/+_Normal mice and other groups) and *, $P < 0.05$; **, $P < 0.01$ (for comparisons between npc1–/–_Normal mice and other groups).

the PCL of the cerebellum (Fig S1S–Z) can promote Purkinje cell degeneration. Previous works failed to detect infiltrating cells in the cerebella of NPC1 mice, possibly due to an increased sensitivity of NPC1 cells to the enzymatic dissociation required for flow cytometry and single-cell transcriptome analyses (Cougnoux et al, 2018a, 2020; Cho et al, 2019). CNS-resident macrophages can be divided into parenchymal microglia and non-parenchymal border–associated macrophages (BAMs). The latter include perivascular, meningeal, and choroid plexus macrophages, and these cells have been originally assumed to derive from peripheral circulating monocytes (Aguzzi et al, 2013; Prinz & Priller, 2014). A recent study by Goldmann et al (2016) has demonstrated that the perivascular and the meningeal macrophages derive from primitive precursors in the yolk sac that are seeded in the brain before birth and are not replenished by blood monocytes in adulthood (Goldmann et al, 2016) like parenchymal microglia (Ginhoux et al, 2010; Schulz et al, 2012; Kierdorf et al, 2013). The perivascular macrophages are potentially involved in angiogenesis, vascular maintenance, and nutrient uptake in healthy brains (Jais et al, 2016; Van Hove et al, 2019). In the patients with MS and in EAE animals, infiltrating macrophages are known to markedly accumulate in the perivascular area (Zhang et al, 2011; Mammana et al, 2018) and in the nodes of Ranvier where they trigger the demyelination of axons (Ajami et al, 2011; Yamasaki et al, 2014; Dendrou et al, 2015). The lack of glia-derived CCL2, a chemokine for CCR2-positive infiltrating monocytes, reduced the recruitment of peripheral macrophages into the CNS and ameliorated the disease score of EAE mice (Dogan et al, 2008; Moreno et al, 2014; Mammana et al, 2018). We speculated that the BM-derived macrophages found in the cerebellar perivascular zones (Figs 2L–O and S1S–Z) may possess distinct properties from the so-called "perivascular macrophages," and may be involved in neuronal degeneration in a disease-specific manner. A neuroinflammatory microenvironment shaped by a modest cell-autonomous Purkinje cell damage (Elrick et al, 2010), potentially induced by defective autophagy (Pacheco & Lieberman, 2008; Sarkar et al, 2013), disturbed calcium homeostasis (Tiscione et al, 2019, 2021), and/or the activation of the STING pathway (Chu et al, 2021), may attract peripheral monocytes/macrophages. Constitutive activation of Toll-like receptor 4 and production of inflammatory cytokines in NPC1 cerebral cells, including astrocytes (Suzuki et al, 2007), may also attract peripheral cells, although these changes may potentially be induced by the infiltrating immune cells. A detailed analysis is necessary to elucidate the brain-derived factors that attract the peripheral innate immune cells in the NPC1 brain.

In ALS patients, the peripheral monocytes have proinflammatory signatures and may play a role in disease progression; in fact, monocytes from rapidly progressing ALS patients had more pronounced proinflammatory gene expression profiles compared with those of slowly progressing ALS patients (Fyfe, 2017; Zhao et al, 2017). The modification of peripheral monocytes by the treatment with an anti-Ly6C antibody or graft of gene-modified BM cells can slow the disease progression and extend the survival of ALS mice (Butovsky et al, 2012; Chiot et al, 2020). In our current study, several genes were found to be specifically up-regulated in NPC1 monocytes, which are involved in the inflammatory process and in cytokine production, thereby potentially promoting Purkinje cell degeneration. Similar

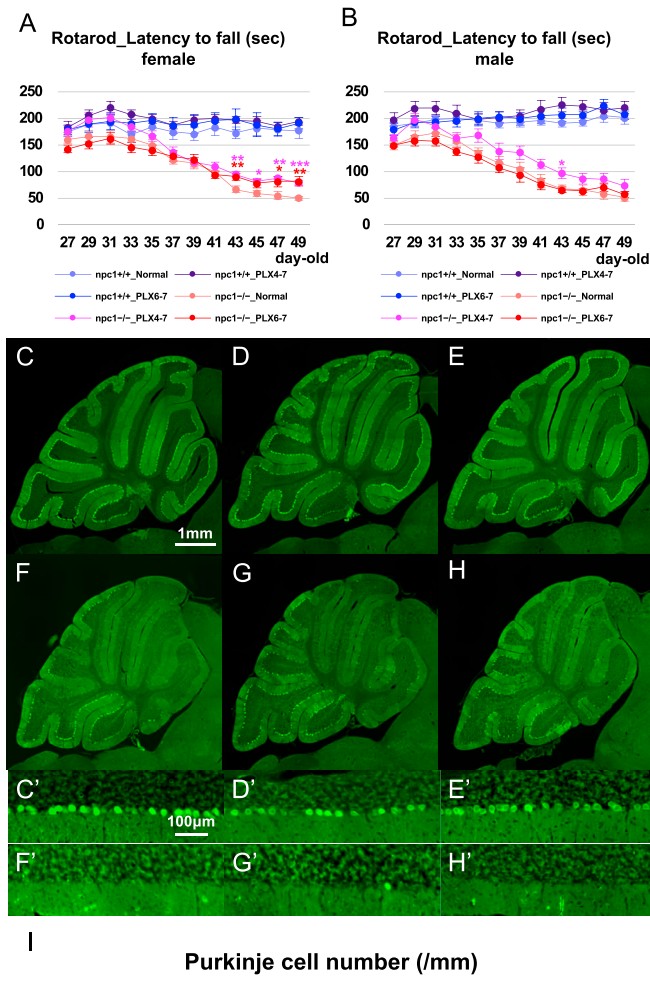

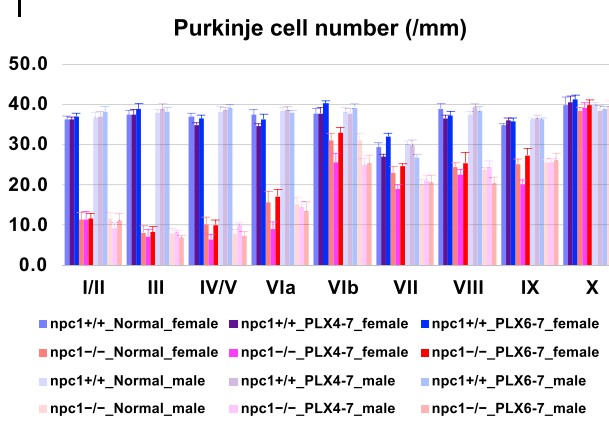

results have been reported previously (Welch et al, 2007; Cologna et al, 2014; Cougnoux et al, 2019). If BM transplantation with modification of disease-related genes is combined with the depletion of the disease-causing monocytes (by targeting the disease-specific cell surface molecule), it may exert a more prominent neuroprotection in NPC1. In this regard, the glycoprotein nonmetastatic melanoma protein B (GPNMB) in NPC1 monocytes (Fig S3E and F and Table S1) may be a suitable target. Recent studies have demonstrated that the GPNMB is selectively up-regulated, and can be a disease-associated biological marker in several neurodegenerative diseases such as NPC1 (Marques et al, 2016; Fukaura et al, 2021; Rodriguez-Gil et al, 2021), Gaucher disease (Zigdon et al, 2015; Kramer et al, 2016), ALS (Tanaka et al, 2012), AD (Hüttenrauch et al, 2018; Aichholzer et al, 2021), and PD (Moloney et al, 2018). GPNMB is a type I transmembrane protein and could play a protective role in neuroinflammation (Huang et al, 2012; Budge et al, 2018; Tsou & Sawalha, 2020; Suda et al, 2022). GPNMB is also specifically expressed in senescent cells, although a vaccination against GPNMB can eliminate these cells and improve age-associated pathologies (Van Deursen, 2014; Suda et al, 2021, 2022). This strategy may be applicable to the treatment of several neurodegenerative diseases (including NPC1) through the elimination of harmful circulating monocytes. In our study, STING pathway (*Tmem173* and *Irf3*) and interferon-stimulated (*Cxcl10*, *Usp18*, and *Ifit1*) genes that can strongly induce Purkinje cell degeneration (Chu et al, 2021) were found unchanged in the peripheral npc1−/− monocytes.

Microglial cells were depleted partially by the gene ablation of *Csf-1* (Figs 7 and 8) or the oral administration of PLX3397 (Figs 9 and 10); however, the Purkinje cell degeneration was not affected in these mouse models. The results are consistent with a previous report (Cougnoux et al, 2020). Like in the case of microglia, the survival of BAMs is dependent on CSF-1R signaling (Mrdjen et al, 2018; Van Hove et al, 2019), thereby suggesting that the simultaneous elimination of these cells did not have a detrimental impact on the survival of Purkinje cells in our model. The *Csf1r*^ΔFIRE/ΔFIRE mice may be useful for the depletion of the parenchymal microglia, but not of non-parenchymal BAMs (Rojo et al, 2019) because of the fact that these mice do not exhibit the neurological and developmental abnormalities reported in *Csf1r*−/− rodents (Patkar et al, 2021). There are several reports on the PLX3397-mediated depletion of microglial cells in mouse models of neurodegenerative diseases. In a 1-methyl-4-phenyl-1,2,3,6-tetrahydropyridine mouse model of

**Figure 10. Non-significant effect on neurodegenerative phenotype with a partial depletion of microglia by oral administration of CSF-1R inhibitor PLX3397.**

**(A, B)** The accelerating rotarod test (accelerated from 0–50 rpm in 300 s) was used, and the latency to fall (in sec) was measured for female (A) and male (B) npc1+/+ _Normal (normal diet), npc1+/+_PLX4-7 (PLX3397 treatment at the ages of 4–7 wk-old), npc1+/+_PLX6-7 (PLX3397 treatment at the ages of 6–7 wk-old), npc1−/− _Normal, npc1−/−_PLX4-7, and npc1−/−_PLX6-7 groups in 27–49-d-old mice. Data are expressed as mean ± SEM (n = 7–14 mice per group in npc1+/+ groups; and n = 8–11 mice per group in npc1−/− groups). The legends for each group are indicated below. For the undertaking of statistical analysis, the two-tailed *t* test was used; *, P < 0.05; **, P < 0.01; ***, P < 0.001 (for comparisons between npc1−/− _Normal mice and other groups). **(C, D, E, F, G, H, C′, D′, E′, F′, G′, H′)** Representative pictures of the immunostaining for calbindin⁺ Purkinje cells in the cerebella of 7-wk-old female npc1+/+ _Normal (C, C′), npc1+/+_PLX4-7 (D, D′), npc1+/+ _PLX6-7 (E, E′),

npc1−/− _Normal (F, F′), npc1−/−_PLX4-7 (G, G′), and npc1−/−_PLX6-7 (H, H′). **(C, D, E, F, G, H, C′, D′, E′, F′, G′, H′)** Enlarged images of panels (C, D, E, F, G, H) are indicated in panels (C′, D′, E′, F′, G′, H′), respectively. **(F, G, H, F′, G′, H′)** Note that Purkinje cells in anterior lobules are lost to a similar extent in npc1−/−_PLX4-7 (G, G′) and npc1−/− _PLX6-7 mice (H, H′) compared with npc1−/−_Normal mice (F, F′). Scale bars in (C): 1 mm (applicable to panels (C, D, E, F, G, H)) and in (C′): 100 µm (applicable to panels (C′, D′, E′, F′, G′, H′)). **(I)** The number of Purkinje cells counted in every lobule of the cerebella of female and male mice. Data are expressed as mean cell number per millimeter (length of Purkinje cell layer) ± SEM (n = 6 mice per group in npc1+/+ groups; and n = 8–10 mice per group in npc1−/− groups). The legends for each group are indicated below. For the undertaking of statistical analysis, one-way ANOVA followed by Tukey HSD post hoc test was used for multiple comparison. No significant differences were identified between npc1−/−_Normal mice and other groups in either gender.

PD, the PLX3397-mediated depletion of microglia exacerbated the observed motor impairment and the dopaminergic neuronal loss, thereby suggesting a protective role of microglia (Yang et al, 2018). Spiller et al (2018) have demonstrated the neuroprotective functions of microglia in an ALS mouse model in which the human TDP-43 can be reversibly induced. In fact, they have shown a robust proliferation of microglia after the suppression of human TDP-43 expression, and these microglial cells have selectively cleared the neuronal human TDP-43 protein (Spiller et al, 2018). However, the PLX3397-mediated elimination of microglia in the early recovery phase has failed to restore the motor function (Spiller et al, 2018). On the other hand, the elimination of microglia in AD mice has resulted into a significant reduction of the Aβ pathology and in an improvement of the cognitive function (Sosna et al, 2018; Son et al, 2020; Delizannis et al, 2021). Microglia may play a beneficial or detrimental role, depending on the disease stage and the neuronal subtypes with which they interact (Guerrero & Sicotte, 2020; Yong, 2022). Cougnoux et al (2020) have shown that a subset of NPC1 microglia can be classified as DAM, supporting their concept that subpopulations of microglia have neurotoxic or neuroprotective activity in NPC1 mice (Cougnoux et al, 2018b, 2020). Keren-Shaul et al (2017) have demonstrated the emergence of DAM by comprehensively analyzing all immune populations in the brain of AD, ALS, and aging mice with the use of single-cell RNA-seq. The differentiation of homeostatic microglia to DAM is a sequential two-step process involving the transition to stage 1 DAM (that depends on unknown signals) and the transition to stage 2 DAM (that depends on the triggering receptor expressed on myeloid cells 2 or TREM2 signaling) (Keren-Shaul et al, 2017; Deczkowska et al, 2018). Human genetic studies have indicated that loss-of-function mutations of the DAM–up-regulated genes and gain-of-function mutations of the DAM–down-regulated genes can increase the risk of developing AD, thereby suggesting that DAM represent a protective response against the AD milieu (Deczkowska et al, 2018). However, there have been conflicting reports on whether the amyloid pathology can be enhanced or be attenuated by TREM2 deficiency in AD mice (Jay et al, 2017; Deczkowska et al, 2018). A recent study has suggested that DAM may also play stage-specific roles in the process of neurodegeneration in AD (Jay et al, 2017). A stage-specific deletion of the TREM2 gene may be important for the analysis of the precise functions of the DAM in neurodegenerative diseases.

In the present study, we investigated an involvement of the peripheral immune cells in the progression of neuronal degeneration in NPC1 mice. We have found that CD4-positive T cells and Tregs can exert a neuroprotective function, and that Ly6C-positive inflammatory monocytes are involved in the progression of disease. However, the depletion of these cells did not affect the survival time of NPC1 mice in our study. The astrocyte-specific overexpression of NPC1 has been shown to triple the lifespan of NPC1 mice (Zhang et al, 2008), thereby suggesting that more comprehensive studies may be required to explore the participation of other CNS-resident and peripheral cells. These efforts will lead to the development of new therapies for this devastating neurodegenerative disease.

# Materials and Methods

### Mice

The experimental protocols were approved by the Ethics Review Committee for Animal Experimentation of the National Center for Child Health and Development. All surgical operations were performed according to rules set forth by the Ethics Committee for the Use of Laboratory Animals at the same center. $Npc1^{nih}$ (BALB/cNctr–$Npc1^{m1N}$/J; strain number: 003092), Rag1-knockout (C.129S7(B6)–$Rag1^{tm1Mom}$/J; 003145), Foxp3–GFP–Tg (C.Cg–$Foxp3^{tm2Tch}$/J; 006769), Foxp3–DTR–Tg (C.B6–Tg(Foxp3–DTR/EGFP)23.2Spar/Mmjax, 011010), Ccr2-RFP/RFP (B6.129(Cg)-$Ccr2^{tm2.1Ifc}$; 017586), and Csf-1op/op mice (B6; C3Fe $a/a$-$Csf1^{op}$/J, 000231) were purchased from the Jackson Laboratory. KuO–Tg mice were provided by Dr. Otsu of the Institute of Medical Science, University of Tokyo (Hamanaka et al, 2013). Ccr2–RFP/RFP, Csf-1op/op, and KuO–Tg mice were backcrossed with a BALB/c genetic background for at least 12 generations. All mice used were maintained on a 99.9% BALB/c genetic background in a 12-h light/12-h dark cycle (standard low light housing conditions), and were supplied with food and water ad libitum. Mating between heterogyzous $Npc1^{nih}$ mice generated mutant and wild-type offspring. Genotyping was performed by using genomic DNA isolated from tail biopsies and the REDExtract-N-Amp Tissue PCR Kit (Merck KGaA). The Foxp3–DTR–Tg and their non-Tg littermates received i.p. injections of diphtheria toxin (10 ng/g) for seven consecutive days when they reached 28 d of age. PLX3397 (pexidartinib; Selleck Biotechnology Co. Ltd), interblended with a powdered chow (290 mg/kg chow; Elmore et al, 2014), was provided freshly every 2–3 d.

### Blood–brain barrier breaching

24 h after the i.p. injection of Evans blue (1% dissolved in PBS; Nacarai Tesque), mice were transcardially perfused with PBS (under deep anesthesia) to remove the dye from the intraluminal space of blood vessels, and then with 4% paraformaldehyde in PBS (Nacarai Tesque). Brains were removed and post-fixed in 4% paraformaldehyde in PBS for 48 h. Sagittal sections of the brain (20 μm in thickness) were mounted on slides by using the Vectashield mounting medium with DAPI (Vector Laboratories Inc.). Mice were transcardially perfused with sulfo-NHS-biotin (0.5 mg/ml in PBS; Thermo Fisher Scientific) followed by 1% paraformaldehyde in PBS (Nacarai Tesque). Brains were removed and post-fixed in 4% paraformaldehyde in PBS for 48 h. The free-floating sagittal sections of the brain (20 μm in thickness) were washed in a PBS medium containing 0.05% Triton X-100 (PBS-T) and were then incubated with DyLight649-conjugated streptavidin (1:50; Vector Laboratories Inc.) in PBS-T, at 4°C, for 2 h. Sections on the slides were mounted by using the Vectashield mounting medium with DAPI (Vector Laboratories Inc.). Images were captured at room temperature with the use of a florescent microscope BIOREVO BZ-9000 system (Keyence) equipped with 10×/0.45-mm objective lens (CFI Plan Apochromat Lambda; Nikon Solutions Co. Ltd). BZ-II Viewer software was used for the acquisition of fluorescent images.

### Neonatal BM–derived cell transplantation

Whole BM cells were obtained by flushing tibiae and femurs of KuO–Tg mice (5–8-wk-old) with ice-cold RPMI 1640 media (Merck KGaA), and mononuclear cells were isolated through a Ficoll-Paque density gradient (PREMIUM 1.084; Cytiva). Neonatal pups (P2–P6) received a single i.p. injection of busulfan (40 $\mu$g/g) followed by an i.p. injection of BM mononuclear cells (1–2 × 10$^6$ cells per pup) at 24 h after the busulfan injection.

### Motor coordination test

The accelerating rotarod test was performed to assess the cerebellar ataxia of mice by using a rotarod treadmill for rats and mice (3-cm diameter rotating cylinder; Model MK-670; Muromachi Kikai Co., Ltd) (Brooks & Dunnett, 2009). The test included three runs (first trial; accelerated from 0 to 50 rpm in 300 s), 10–15-min interval, and three runs (second trial; the same accelerating speed). The training session was performed on 25-d-old mice, whereas the latency to fall (in seconds) was measured every 2 d (starting from 27-d-old mice) and was calculated as the average value of the three runs of the second trial.

### Antibodies and immunohistochemistry

The primary antibodies used in this study were as follows: mouse anti-calbindin (clone CB-955; diluted at 1:2,000; Merck KGaA), rabbit anti-Iba1 (1:1,000; FUJIFILM Wako Pure Chemical Corporation), goat anti-Iba1 (1:100; Abcam plc), rabbit anti-Glut-1 (1:100; Merck KGaA), rat anti-CD11b (clone 5C6; 1:20; BioRad Laboratories Inc.), rat anti-CD31 (clone MEC 13.3; 1:20; BD Biosciences), mouse anti-laminin (clone LAM-89; 1:1,000; Merck KGaA), rabbit anti-collagen type IV (1:400; Abcam plc), and rabbit anti-Tmem119 (1:200; Abcam plc). The free-floating sagittal sections of the mouse brain (20 $\mu$m in thickness) were washed in PBS-T, were soaked with mouse Ig blocking reagent (Vector M.O.M. Immunodetection Kit; Vector Laboratories Inc.) in PBS-T for 1 h, and were then incubated with the primary antibodies dissolved in PBS-T containing the M.O.M. protein concentrate (Vector M.O.M. Immunodetection Kit; Vector Laboratories Inc.) at 4°C for 48 h. Subsequently, the sections were incubated for 2 h in the same fresh medium containing the following secondary antibodies: AMCA-, FITC-, or Cy5-conjugated anti-mouse IgG, AMCA-, FITC-, or Cy5-conjugated anti-rabbit IgG, FITC-conjugated anti-rat IgG, and Cy5-conjugated anti-goat IgG antibodies (1:400; Jackson ImmunoResearch Laboratories, Inc.). Sections on the slides were mounted by using the Vectashield mounting medium with or without DAPI (Vector Laboratories Inc.). Images were captured at room temperature by using a florescence microscope BIOREVO BZ-9000 system (Keyence) or a confocal laser–scanning microscope IX83-FV3000 system (Olympus Scientific Solutions). The florescence microscope BIOREVO BZ-9000 system was equipped with 10×/0.45-mm objective lens (CFI Plan Apochromat Lambda, Nikon Solutions Co. Ltd). The BZ-II Viewer software was used for the acquisition of fluorescent images, whereas the BZ-II Analyzer software was used for the automated image stitching. The confocal laser–scanning microscope IX83-FV3000 system was equipped with 60×/0.11-mm

objective lens (UPLAPO60XOHR; Olympus Scientific Solutions) and 100×/0.12-mm objective lens (UPLAPO100XOHR; Olympus Scientific Solutions). The FV31S-SW software was used for the acquisition of single slices and 3D projection images. Z-stacks were acquired at 1.02-$\mu$m intervals (17 slices per image).

### Cell count

The number of Purkinje cells was counted, and the length of the PCL was measured in every lobule of the cerebellum by using the BZ-II Analyzer software (Keyence). Data were expressed as the mean cell number per millimeter (length of PCL) ± SEM in every lobule. The number of Iba1+ cells was counted, and the areas of the ML and the PCL (designated as ML/PCL) or the granule cell layer (GCL) and the white matter (WM; designated as GCL/WM) were measured in every lobule of the cerebellum by using the BZ-II Analyzer software (Keyence). Data were expressed as the mean cell number per square millimeter (area of ML/PCL or GCL/WM) ± SEM.

### Flow cytometry and FACS

The peripheral blood of mice was treated with eBioscience RBC lysis buffer (Thermo Fisher Scientific) followed by an anti-CD16/32 antibody (clone 93; BioLegend Inc.). Immune cells were analyzed by a flow cytometry BD LSR Fortessa (BD Biosciences) by using the following antibodies: FITC- or PE-conjugated anti-CD45.2 (clone 104; BioLegend Inc.), APC-conjugated anti-CD11b (clone M1/70; BioLegend Inc.), FITC-conjugated anti-CD11b (clone M1/70; BD Biosciences), PE-Cy7–conjugated anti-Ly6C (clone AL-21; BD Biosciences), BV421-conjugated anti-CD4 (clone GK1.5; BioLegend Inc.), PerCP/Cy5.5-conjugated anti-CD25 (clone PC61; BioLegend Inc.), and APC-conjugated anti-CD25 (clone PC61; BD Biosciences) antibodies. The peripheral monocytes for RNA-seq and T-cell populations for the infusion experiment were collected by BD FACS Aria III and BD FACS Melody (BD Biosciences), respectively, by using the following antibodies: FITC-conjugated anti-CD45.2 (clone 104; BioLegend Inc.), APC/Cy7-conjugated anti-CD11b (clone M1/70; BioLegend Inc.), APC-conjugated anti-CD3e (clone 145-2C11; Beckman Coulter Inc.), FITC-, APC/Cy7-, or BV421-conjugated anti-CD4 (clone GK1.5; BioLegend Inc.), APC/Cy7-conjugated anti-CD8a (clone 53-6.7; BioLegend Inc.), APC-conjugated anti-CXCR3 (clone CXCR3-173; BD Biosciences), BV421-conjugated anti-CCR4 (clone 2G12; BioLegend Inc.), PE-conjugated anti-CCR6 (clone 29-2L17; BioLegend Inc.), and APC-conjugated anti-CD25 antibodies (clone PC61; BD Biosciences).

### RNA-seq

Total RNA was purified by using the RNeasy Plus Mini Kit according to the manufacturer's manual (QIAGEN NV). The mouse monocyte RNA was assayed as an integrity quality control by using the RNA 6000 Pico kit (Agilent Technologies, Inc.) on the Agilent 2100 Bioanalyzer (Agilent Technologies, Inc.). RNA samples with an RNA integrity number of at least 6.9 were subjected to deep RNA-seq. Sequencing libraries were prepared with the use of the Ovation SoLo RNA-Seq library preparation kit (Tecan Group Ltd) by using 10-ng total RNA. Libraries were sequenced by using the Illumina NextSeq (Illumina, Inc.). Gene transcripts were quantified by using a tool salmon (Patro

et al, 2017) and were analyzed by EdgeR (Robinson et al, 2010). The GO analyses were performed through webtools g:Profiler (https://biit.cs.ut.ee/gprofiler/gost; [Reimand et al, 2016]) and ShinyGO (http://bioinformatics.sdstate.edu/go/; [Ge et al, 2020]).

## Supplementary Information

## Acknowledgements

This work was supported by JSPS KAKENHI (grant numbers: JP16K10006 and JP19K08266). We are grateful to Dr Hideki Tsumura and Ms Miyuki Shindo (of the Division of Laboratory Animal Resources at the National Institute for Child Health and Development) for their devoted support in the undertaking of our animal experiments.

### Author Contributions

T Yasuda: conceptualization, data curation, formal analysis, funding acquisition, validation, investigation, visualization, methodology, project administration, and writing—original draft, review, and editing.
T Uchiyama: supervision, methodology, and project administration.
N Watanabe: investigation.
N Ito: data curation, formal analysis, validation, investigation, and methodology.
K Nakabayashi: software, formal analysis, validation, and methodology.
H Mochizuki: supervision and methodology.
M Onodera: conceptualization, supervision, methodology, and project administration.

### Conflict of Interest Statement

The authors declare that they have no conflict of interest.

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
