## [Reviewer comments · Life Science Alliance]

Peripheral immune system modulates Purkinje cell degeneration in Niemann-Pick disease type C1

Toru Yasuda, Toru Uchiyama, Nobuyuki Watanabe, Noriko Ito, Kazuhiko Nakabayashi, Hideki Mochizuki, and Masafumi Onodera

DOI: <https://doi.org/10.26508/lsa.202201881>

Corresponding author(s): Toru Yasuda, National Center For Child Health and Development

Review Timeline:	Submission Date:	2022-12-19
	Editorial Decision:	2023-02-20
	Revision Received:	2023-05-21
	Editorial Decision:	2023-06-09
	Revision Received:	2023-06-13
	Accepted:	2023-06-14

Transaction Report:

February 20, 2023

Re: Life Science Alliance manuscript #LSA-2022-01881-T

Dr. Toru Yasuda
National Center For Child Health and Development
Department of Human Genetics
2-10-1 Okura
Setagaya-ku
Tokyo 157-8535
Japan

Dear Dr. Yasuda,

Thank you for submitting your manuscript entitled "Peripheral immune system modulates Purkinje cell degeneration in Niemann-Pick disease type C1" to Life Science Alliance. The manuscript was assessed by expert reviewers, whose comments are appended to this letter. We invite you to submit a revised manuscript addressing the Reviewer comments.

Thank you for this interesting contribution to Life Science Alliance. We are looking forward to receiving your revised manuscript.

Sincerely,

B. MANUSCRIPT ORGANIZATION AND FORMATTING:

Reviewer #1 (Comments to the Authors (Required)):

Here Yasuda et.al. present a set of experiments investigating the impact of peripheral and resident immune cells in the time-course of Purkinje cell degeneration in NPC1 mice. Overall, the experiments appear to be well executed, and the presented data match the conclusions drawn in the text. While none of the interventions performed had an impact on survival time of the NPC1 mice - several had impacts on the initiation of Purkinje cell degeneration and ataxic phenotypes, adding novel mechanistic insight into NPC1 disease initiation, which could lead to helpful therapeutic candidates.

Issues that the authors need to address to improve the manuscript are as follows:

- 1) A common theme throughout several of the experiments performed, was a significant difference in responses from male and female mice, with female mice consistently more responsive to immune cell modulation. Given this sex effect, and the observation that only NPC1 mice contained infiltrating immune cells from KuO transplantation, did the authors look for any sex differences in BBB leakage and immune cell infiltration? Figure legends for Fig.2 N-O and Fig. S1 do not mention sex of the animals imaged. This comparison would add significance and mechanism to the lack of effect in male mice from BM-transplantation, Rag1-KO, and CD4+ cell infusion. However, if male mice also show BBB immune infiltration - authors should speculate on a sex-specific mechanism that explains this discrepancy in the text.
- 2) In regard to RNA-seq performed on peripheral monocytes, it would also add critical insight into this sex-specific response if male monocytes were also analyzed - which is unfortunately lacking here. For the female samples analyzed, did the authors perform a background correction for Gene Ontology analysis? Given that the gene expression signature in these cells will be heavily enriched for immune related processes, it is critical to correct the DEG enrichment according to the background gene-set detected in these cells. A different GO tool such as ShinyGO should be used to include a background correction.

Minor Comments

Page 18, paragraph 2 - "NCP1" should read "NPC1"

Reviewer #2 (Comments to the Authors (Required)):

The present study entitled "Peripheral immune system modulates Purkinje cell degeneration in Niemann-Pick disease type C1" by Yasuda and coworkers is an interesting study building up on the existing literature. The authors validate existing observations, the lack of effect of Csfr1 inhibition on survival, the transcriptomic changes in myeloid cells following the loss of Npc1. The authors also show that the timing of the bone marrow transplant impacts the outcome of the treatment. More importantly the present work investigates the potential role of lymphocytes in the disease and most notably Treg that have received little attention in NPC. The authors point that targeting T cell biology might be a promising avenue in NPC1 and this deserves further investigations. It is surprising that the authors performed the transcriptomic analysis on the monocytes rather than the T cells but this opens the way for more in depth studies of these cells in the future. It is worth mentioning that the authors studied separately the effect in males and female animals. This is especially relevant to NPC1 where a gender difference in the survival, of mice, has been reported by independent groups. Overall, the manuscript is well written, the experiments well designed, and the results clearly presented.

While good and fitting to the target journal I have a few comments and suggestions:

- The BBB integrity in the same mouse model was previously evaluated using a different experimental setting and no leakage was observed. Could the discrepancy in the results come from the sensitivity of the method used?
- The neonatal BM transplant is effective while similar transplantation in older animal is not. This is an interesting observation that should be discussed further especially the discrepancies of results depending on the age of the animals at transplant. What

could be the reasons of such defenses should be elaborated?

- Several studies using different methodological approaches (<https://academic.oup.com/hmg/article/27/12/2076/4956805> ; <https://onlinelibrary.wiley.com/doi/full/10.1111/jnc.14483> ; <https://www.ncbi.nlm.nih.gov/pmc/articles/PMC7432835/>) failed to identify infiltrating immune cells in the cerebellum. The discrepancy between the authors results and the literature should be discussed.
- The lack of effect of PLX3397 treatment has previously been reported (<https://doi.org/10.3390/ijms21155368>) the results were therefore to be expected. In the same study the similarities with the "DAM" signatures from other mouse model were evaluated and discussed.
- Newton and coworker (<https://www.ncbi.nlm.nih.gov/pmc/articles/PMC5349795/>) evaluated the effect of FTY720 in NPC. In this study the authors suggested it may be a promising avenue in NPC. Based on your results lymphodepletion would exacerbate the disease. These maybe conflicting observations should be discussed?
- From the genes up regulated in the monocytes and suggested to be involved in the pathology several have already been studied and could be included in the discussion or at least referenced: (<https://www.ahajournals.org/doi/10.1161/CIRCULATIONAHA.107.701276> ; <https://www.ncbi.nlm.nih.gov/pmc/articles/PMC3877698/> ; <https://www.ncbi.nlm.nih.gov/pmc/articles/PMC6935239/>)
- TLR4 was found constitutively activated in NPC (<https://www.jneurosci.org/content/27/8/1879>) it is surprising not to find any signature of this mechanism in your analysis. Do you have any hypothesis regarding this?
- It might be more accurate to describe the rotarod test as a motor coordination test rather than a behavior test.
- The reference for g:Profiler is missing (<https://pubmed.ncbi.nlm.nih.gov/27098042/>)
- The reference for EdgeR is missing (<https://pubmed.ncbi.nlm.nih.gov/19910308/>)
- Several reports, including in NPC, have shown that Gapdh shouldn't be used as internal control for myeloid cells (e.g. <https://bmciimmunol.biomedcentral.com/articles/10.1186/1471-2172-11-21>). Why were the genes expression normalized to Gapdh? This has to be justified or additional experiments demonstrating that Gapdh has a stable expression between control and mutant cells should be performed or at least evaluated in the transcriptomic data.

Reviewer #3 (Comments to the Authors (Required)):

This is an interesting manuscript that reports on the significance of peripheral immune cells on CNS neurodegeneration in a mouse model of Niemann-Pick disease type C1 (NPC1), which is a lysosomal storage disorder among others characterized by progressive neuronal degeneration. Manipulation of the immune system may have perspectives as a novel therapeutic adjacent to current lipid reductions. The manuscript describes how bone marrow-derived CD4-positive cells transplantation partly quite dramatically ameliorates Purkinje cell degeneration and restore functions in NPC1 mice. Interestingly mature lymphocytes, next to transplanted cells, also have neuroprotective effects. On the reverse depletion of T cells promoted neurodegeneration. Circulatory monocytes also promoted neurodegeneration. Overall, the study suggests a protective role of the specific immune system and the reverse of the innate ditto.

The authors should take these comments into consideration:

Protocol: What is meant by BBB breaching? Blood-brain barrier? Please explain

A concern is the illustrations underpinning the FACS and survival data: Morphological evidence is provided for microglia infiltration; but there is not much evidence for lymphocyte infiltration or macrophage infiltration. The demonstration of perivascular cells does not really convincingly show direct interaction between Purkinje cells and cells of the immune system. Please explain

The comments on the blood-brain barrier in the supplementary are not sufficient. There must be a better provision of the criteria for BBB breaching and the broad staining by Evans Blue is almost too good. The animals must be very sick if this is true BBB breaching; does this lead to labeling of Purkinje cells? Does this breaching lead to extravasation and presence of immunoglobulins in Purkinje cells?

Dear Dr. Eric Sawey,
Executive Editor,
Life Science Alliance

Manuscript: LSA-2022-01881-T, entitled “Peripheral immune system modulates Purkinje cell degeneration in Niemann-Pick disease type C1”

Authors: Yasuda T et al.

Dear Dr. Eric Sawey,

Thank you very much for your review and comments for our manuscript. The comments raised by the reviewers were very helpful to improve our manuscript. According to their comments, we newly performed several experiments and revised the manuscript to meet the comments. According to your request, we prepared our point-by-point responses to the reviewers' comments, which were raised in the reviewing process of Life Science Alliance.

The changes made in the manuscript and our point-by-point responses are provided below:

Responses to the comments by Reviewer #1

1) A common theme throughout several of the experiments performed, was a significant difference in responses from male and female mice, with female mice consistently more

responsive to immune cell modulation. Given this sex effect, and the observation that only NPC1 mice contained infiltrating immune cells from KuO transplantation, did the authors look for any sex differences in BBB leakage and immune cell infiltration? Figure legends for Fig.2 N-O and Fig. S1 do not mention sex of the animals imaged. This comparison would add significance and mechanism to the lack of effect in male mice from BM-transplantation, Rag1-KO, and CD4+ cell infusion. However, if male mice also show BBB immune infiltration - authors should speculate on a sex-specific mechanism that explains this discrepancy in the text.

We will deeply appreciate the reviewer's excellent comment to improve our manuscript. We have investigated the BBB leakage and infiltration of KuO-positive immune cells in both female and male mice and have not found any sex differences. All images shown in Fig.2 N-O and Fig.S1 were from female mice. Accordingly, we have revised the description that reviewer pointed out as shown below:

Page 10, lines 173-177 in the revised manuscript;

Results section

(line 176, **shown in red**, have been newly added to the previously submitted manuscript);

The histological analysis of the KuO-positive cells in the cerebellum revealed an invasion of the KuO-CD11b-double-positive monocyte-derived macrophages into the perivascular zone of the interlobular space of the cerebellum **in both female and male mice** (Fig. 2 M–O and Fig. S1).

Page 11, line 192 in the revised manuscript;

Results section

We did not find any sex differences in these experiments.

Page 84, lines 1491-1493 in the revised manuscript;

Figure legends section

(N and O) Confocal images of the KuO⁺ BM-derived cells. Sagittal sections of **female** mouse cerebella were immunostained for monocytes/macrophages (anti-CD11b, green) and blood vessels (anti-Glut-1, gray).

Page 1, lines 5-7 in the revised Supplemental figure legends;

(A–R) Sagittal sections of **female** mouse cerebella were immunostained for monocytes/macrophages (anti-CD11b; green) and blood vessels (anti-Glut-1; gray).

We newly carried out bone marrow transplantation using *npc1^{-/-}* KuO-transgenic mice as a donor. As shown in the revised Supplemental figure 1 S-Z, we found an invasion of the KuO-CD11b-double-positive monocyte-derived macrophages not only into the perivascular zone of the interlobular space, but also into the perivascular zones of molecular layer and Purkinje cell layer of the cerebellum. We speculated that these results strengthen our concept that peripheral immune cells can be involved in the modulation of the neurodegenerative process of NPC1. Accordingly, we have revised our manuscript to provide the new data as shown below:

Revised Supplemental figure 1 S-Z

Pages 1-2, lines 12-27 in the revised Supplemental figure legends;

(S and T) Immunohistochemical analysis for female *npc1*^{+/+} BMT (S) and *npc1*^{-/-} BMT mice (T) using *npc1*^{-/-} KuO-Tg mice as a donor. Sagittal sections of mouse cerebella were immunostained for Purkinje cells (anti-Calbindin, green) and microglia/macrophages (anti-Iba1, red). Note that the KuO⁺ BM-derived cells (blue) were localized at the interlobular space (indicated by white arrowheads in panel T) and parenchyma of the cerebellum (blue arrowheads in panel T). Scale bar in (S): 500 µm (applicable to panels S and T) **(U–Z)** Confocal images of the KuO⁺ BM-derived cells. Sagittal sections of female mouse cerebella were immunostained for monocytes/macrophages (anti-CD11b, green) and blood vessels (anti-Glut-1, gray). An enlarged images of panel U, W, and Y are indicated in panels V, X, and Z, respectively. Z-stacks were captured at 1.00-µm intervals by using a confocal laser-scanning microscope. Note that the KuO⁺ BM-derived cells (red) were CD11b⁺ macrophages and localized perivascular zones of the interlobular space (indicated by white arrowhead in panels U and W), the ML (blue arrowhead in panel W), and the PCL of the cerebellum (blue arrowhead in panel Y). Scale bars in (U): 50µm (applicable to panels W and Y) and (V): 10µm (applicable to panels X and Z).

Page 10, lines 178-182 in the revised manuscript;

Results section

When the BM cell transplantation was performed using *npc1*^{-/-} *KuO*-Tg mice as a donor, we observed an invasion of the *KuO*-*CD11b*-double-positive monocyte-derived macrophages into the perivascular zones of the interlobular space, the molecular layer (ML), and the Purkinje cell layer (PCL) of the cerebellum in both female and male mice (Fig. S1 S–Z).

Page 13, line 256 in the revised manuscript;

Results section

may have infiltrated the CNS (Fig. 2 L–O and Fig. S1 S–Z)

Page 23, lines 453-456 in the revised manuscript;

Discussion section

This finding suggests that macrophages infiltrating into the perivascular zones of the interlobular space (Fig. 2 L–O and Fig. S1 S–Z), the ML, and the PCL of the cerebellum (Fig. S1 S–Z) can promote Purkinje cell degeneration.

Page 24, lines 477-480 in the revised manuscript;

Discussion section

We speculated that the BM-derived macrophages found in the cerebellar perivascular zones (Fig. 2 L–O and Fig. S1 S–Z) may possess distinct properties from the so-called “perivascular macrophages,” and may be involved in neuronal degeneration in a disease-specific manner.

Page 32, line 675 in the revised manuscript;

Materials and methods section

the length of the PCL

Page 32, line 678 in the revised manuscript;
Materials and methods section
the areas of the ML

The following sentence was removed.

Page 24, line 490-491 in the revised manuscript;
Discussion section

Moreover, it may be of critical importance to elucidate the distinct localization and fate of the infiltrating macrophages deriving from *npc1*^{-/-} BM cells.

We will agree totally to the reviewer's suggestion that a sex-specific mechanism should be discussed precisely to explain the discrepancy in the BM-transplantation, Rag1-KO, and CD4⁺ cell infusion experiments.

We have carefully read several papers, which showed the effects of sex hormones on adaptive immune cells. In males, androgens show a suppressive effect on lymphopoiesis in early development. While females are known to have higher numbers of total and activated CD4-positive T cells than males, estrogen and progesterone have ability to augment expression of Foxp3 in CD4-positive cells in mice and humans. We speculated these effects can explain the gender-specific neuronal protection and degeneration observed in our present study. And, we will greatly appreciate the reviewer if he/she could check whether the revised manuscript could discuss the gender-specific effects precisely:

Page 20, lines 407-417 in the revised manuscript;
Discussion section

Previous reports showed the effects of sex hormones on adaptive immune cells (Dodd & Menon, 2022; Klein & Flanagan, 2016). Deprivation of androgen and knockout of androgen receptors in mice resulted in thymic enlargement, indicating that androgens have a suppressive effect on lymphopoiesis in early development (Dodd & Menon, 2022; Lai et al., 2012; Roden et al., 2004). Females have higher numbers of total and activated

CD4-positive T cells than males (Abdullah et al., 2012; Klein & Flanagan, 2016; Sankaran-Walters et al., 2013). Moreover, estrogen and progesterone have ability to augment expression of Foxp3 in CD4-positive cells in mice and humans (Lee et al., 2011; Polanczyk et al., 2004). These immune-modulating effects of sex hormones may potentially be responsible for the gender-specific neuronal protection and degeneration observed in this study.

References newly cited in the revised manuscript:

- Abdullah, M., Chai, P. S., Chong, M. Y., Tohit, E. R. M., Ramasamy, R., Pei, C. P., & Vidyadaran, S. (2012). Gender effect on in vitro lymphocyte subset levels of healthy individuals. *Cellular Immunology*, 272(2), 214–219. <https://doi.org/10.1016/J.CELLIMM.2011.10.009>
- Dodd, K. C., & Menon, M. (2022). Sex bias in lymphocytes: Implications for autoimmune diseases. *Frontiers in Immunology*, 13, 945762. <https://doi.org/10.3389/fimmu.2022.945762>
- Klein, S. L., & Flanagan, K. L. (2016). Sex differences in immune responses. *Nature Reviews Immunology*, 16(10), 626–638. <https://doi.org/10.1038/nri.2016.90>
- Lai, J. J., Lai, K. P., Zeng, W., Chuang, K. H., Altuwaijri, S., & Chang, C. (2012). Androgen receptor influences on body defense system via modulation of innate and adaptive immune systems: Lessons from conditional AR knockout mice. *American Journal of Pathology*, 181(5), 1504–1512. <https://doi.org/10.1016/j.ajpath.2012.07.008>
- Lee, J. H., Ulrich, B., Cho, J., Park, J., & Kim, C. H. (2011). Progesterone Promotes Differentiation of Human Cord Blood Fetal T Cells into T Regulatory Cells but Suppresses Their Differentiation into Th17 Cells. *The Journal of Immunology*, 187(4), 1778–1787. <https://doi.org/10.4049/jimmunol.1003919>
- Polanczyk, M. J., Carson, B. D., Subramanian, S., Afentoulis, M., Vandembark, A. A., Ziegler, S. F., & Offner, H. (2004). Cutting Edge: Estrogen Drives Expansion of the

CD4+CD25+ Regulatory T Cell Compartment. *The Journal of Immunology*, 173(4), 2227–2230. <https://doi.org/10.4049/jimmunol.173.4.2227>

Roden, A. C., Moser, M. T., Tri, S. D., Mercader, M., Kuntz, S. M., Dong, H., Hurwitz, A. A., McKean, D. J., Celis, E., Leibovich, B. C., Allison, J. P., & Kwon, E. D. (2004). Augmentation of T Cell Levels and Responses Induced by Androgen Deprivation. *The Journal of Immunology*, 173(10), 6098–6108. <https://doi.org/10.4049/jimmunol.173.10.6098>

Sankaran-Walters, S., Macal, M., Grishina, I., Nagy, L., Goulart, L., Coolidge, K., Li, J., Fenton, A., Williams, T., Miller, M. K., Flamm, J., Prindiville, T., George, M., & Dandekar, S. (2013). Sex differences matter in the gut: Effect on mucosal immune activation and inflammation. *Biology of Sex Differences*, 4(1), 10. <https://doi.org/10.1186/2042-6410-4-10>

2) In regard to RNA-seq performed on peripheral monocytes, it would also add critical insight into this sex-specific response if male monocytes were also analyzed - which is unfortunately lacking here. For the female samples analyzed, did the authors perform a background correction for Gene Ontology analysis? Given that the gene expression signature in these cells will be heavily enriched for immune related processes, it is critical to correct the DEG enrichment according to the background gene-set detected in these cells. A different GO tool such as ShinyGO should be used to include a background correction.

We are grateful to the reviewer for his/her helpful comment. According to the reviewer's comment, we carried out Gene Ontology analysis using a different tool ShinyGO. Several important biological pathways were enriched by the ShinyGO analysis, including steroid biosynthesis, cholesterol metabolism, and cytokine-cytokine receptor interaction. We have revised our manuscript to provide the data in Table S2 based on our idea described below:

We speculated that the GO data obtained from g:Profiler analysis that have been provided in Table S1 in the previously submitted manuscript may also be informative for

the readers, since this analysis highlighted several critical processes including cell migration, inflammatory response, and the regulation of cytokine production, that will explain our concept that the invasion of peripheral monocytes/macrophages into the cerebellum can be an important process to induce neuronal degeneration. Accordingly, we have revised our manuscript to add the new data from ShinyGO analysis as shown below:

Page 15, lines 298-302 in the revised manuscript;

Results section

We further performed GO analysis using a different tool ShinyGO (Ge et al., 2020). We found several critical biological processes including steroid biosynthesis (enrichment FDR: 2.43e-4), cholesterol metabolism (1.56e-2), and cytokine-cytokine receptor interaction (6.30e-3) were enriched (Table S2).

Page 34, lines 717-719 in the revised manuscript;

Materials and methods section

The GO analyses were performed through webtools g:Profiler (<https://biit.cs.ut.ee/gprofiler/gost>; (Reimand et al., 2016)) and ShinyGO (<http://bioinformatics.sdstate.edu/go/>; (Ge et al., 2020)).

References newly cited in the revised manuscript:

Ge, S. X., Jung, D., Jung, D., & Yao, R. (2020). ShinyGO: A graphical gene-set enrichment tool for animals and plants. *Bioinformatics*, 36(8), 2628–2629.
<https://doi.org/10.1093/bioinformatics/btz931>

Minor Comments

Page 18, paragraph 2 - "NCPI" should read "NPC1"

We appreciate the reviewer's comment. Accordingly, we have revised the description that reviewer pointed out as shown below:

Page 21, line 423 in the revised manuscript;

Discussion section

NPC1 patients.

Responses to the comments by Reviewer #2

- The BBB integrity in the same mouse model was previously evaluated using a different experimental setting and no leakage was observed. Could the discrepancy in the results comes from the sensitivity of the method used?

We are grateful to the reviewer for his/her helpful comments to improve our manuscript. We have carefully read the paper by Cougnoux et al. (Microglia activation in Niemann–Pick disease, type C1 is amendable to therapeutic intervention; Human Molecular Genetics, 2018, Vol. 27, No. 12 2076–2089) that has evaluated BBB integrity using Evans blue dye in the same mouse model. These authors have measured the absorbance of Evans blue dye that is leaked to the brain tissue and normalized to tissue mass. We speculated that this biochemical method may have low sensitivity to detect the extravasated dye. We could detect a little amount of dye histologically, possibly due to the sensitivity of method as the reviewer has pointed out. Accordingly, we have revised our manuscript to describe the result in a previous work, as shown below:

Pages 10-11, lines 186-190 in the revised manuscript

Results section

(lines 188-190, shown in red, have been newly added to the previously submitted manuscript);

Breaching of the BBB was confirmed histologically by a leakage of peripherally injected Evans blue dye and sulfo-NHS-biotin molecules into the parenchyma of the cerebellum of NPC1 mice (Fig. S1 A'–D'), that could not be detected by a biochemical method in a previous work (Cougnoux et al., 2018a).

References revised in the revised manuscript:

The report by Cougnoux et al. (2018) has been revised as Cougnoux et al. (2018a) as follows:

Cougnoux, A., Drummond, R. A., Collar, A. L., Iben, J. R., Salman, A., Westgarth, H., Wassif, C. A., Cawley, N. X., Farhat, N. Y., Ozato, K., Lionakis, M. S., & Porter, F. D. (2018a). Microglia activation in Niemann-Pick disease, type C1 is amendable to therapeutic intervention. *Human Molecular Genetics*, 27(12), 2076–2089.
<https://doi.org/10.1093/hmg/ddy112>

- The neonatal BM transplant is effective while similar transplantation in older animal is not. This is an interesting observation that should be discussed further especially the discrepancies of results depending on the age of the animals at transplant. What could be the reasons of such defenses should be elaborated?

We will deeply appreciate the reviewer's excellent comment to improve our manuscript. We agree to the reviewer's suggestion that the discrepancy of neuroprotective effect depending on the age of the animals at transplant should be discussed further. We have carefully read the paper by Cougnoux et al. (Microglia activation in Niemann–Pick disease, type C1 is amendable to therapeutic intervention; *Human Molecular Genetics*, 2018, Vol. 27, No. 12, 2076–2089) that has evaluated the effect of bone marrow transplantation in the same mouse model. These authors used 3- to 4-week-old wild-type and mutant mice as a recipient. A previous work by Colombo et al. (Loss of NPC1 enhances phagocytic uptake

and impairs lipid trafficking in microglia; Nature Communications, 2021, 12, 1158) has indicated that brain-resident microglia were already activated at pre-symptomatic stage, 7-day-old, while the Calbindin-positive Purkinje cells were preserved intact. Accordingly, we speculated that such microglial state switch may lead to production of inflammatory cytokines and chemokines that may stimulate and recruit peripheral immune cells. We also speculated that a lethal irradiation in their transplantation study might have affected the inflammatory status of peripheral immune cells. Accordingly, we have added several sentences to discuss these important points, as shown below. And, we will greatly appreciate the reviewer if he/she could check whether the revised manuscript could discuss the discrepancy of the neuroprotective effect precisely:

Page 18, lines 368-374 in the revised manuscript;

Discussion section

A previous study has found no change in motor activity, weight loss, or survival in BM-derived cell transplantation at 3–4-wk-old (Cognoux et al., 2018a). Based on the finding that the brain-resident microglia were activated at 7-day-old (Colombo et al., 2021), we speculated that the age of recipients and/or lethal irradiation in their transplantation experiment might have affected the inflammatory state of peripheral immune cells.

- Several studies using different methodological approaches (<https://academic.oup.com/hmg/article/27/12/2076/4956805> ; <https://onlinelibrary.wiley.com/doi/full/10.1111/jnc.14483> ; <https://www.ncbi.nlm.nih.gov/pmc/articles/PMC7432835/>) failed to identify infiltrating immune cells in the cerebellum. The discrepancy between the authors results and the literature should be discussed.

We will deeply appreciate the reviewer's excellent comment. We will agree totally to the comment that the discrepancy between our study and the literatures should be discussed. We have carefully read the following three papers.

- 1) Cougnoux at al. (Microglia activation in Niemann–Pick disease, type C1 is amendable to therapeutic intervention; *Human Molecular Genetics*, 2018, Vol. 27, No. 12, 2076–2089)
- 2) Cho et al. (Absence of infiltrating peripheral myeloid cells in the brains of mouse models of lysosomal storage disorders; *Journal of Neurochemistry*, 2019, 148(5), 625–638)
- 3) Cougnoux at al. (Single Cell Transcriptome Analysis of Niemann–Pick Disease, Type C1 Cerebella; *International Journal of Molecular Sciences*, 2020, 21(15), 5368)

To analyze infiltrating immune cells in the cerebellum, Cougnoux at al. (2018) and Cho et al. (2019) performed flow cytometry, and Cougnoux at al. (2020) performed single cell transcriptome analysis. As described in the paper by Cougnoux at al. (2020), *npc1*^{-/-} cells may have increased sensitivity to the enzymatic disassociation required to prepare samples for flow cytometry and single cell transcriptome analysis. We also speculated that a small number of infiltrating cells in NPC1 were difficult to detect when compared with those in experimental autoimmune encephalomyelitis model (Cho et al. (2019)) or mucopolipidosis type IV model (Cougnoux at al. (2018)). Accordingly, we have revised the description that reviewer pointed out as shown below:

Page 23, lines 456-459 in the revised manuscript;

Discussion section

Previous works failed to detect infiltrating cells in the cerebella of NPC1 mice, possibly due to an increased sensitivity of NPC1 cells to the enzymatic dissociation required for

flow cytometry and single cell transcriptome analyses (Cho et al., 2019; Cougnoux et al., 2018a, 2020).

References newly cited in the revised manuscript:

- Cho, S. M., Vardi, A., Platt, N., & Futerman, A. H. (2019). Absence of infiltrating peripheral myeloid cells in the brains of mouse models of lysosomal storage disorders. *Journal of Neurochemistry*, 148(5), 625–638. <https://doi.org/10.1111/jnc.14483>
- Cougnoux, A., Yerger, J. C., Fellmeth, M., Serra-Vinardell, J., Martin, K., Navid, F., Iben, J., Wassif, C. A., Cawley, N. X., & Porter, F. D. (2020). Single cell transcriptome analysis of niemann–pick disease, type c1 cerebella. *International Journal of Molecular Sciences*, 21(15), 5368. <https://doi.org/10.3390/ijms21155368>

- The lack of effect of PLX3397 treatment has previously been reported (<https://doi.org/10.3390/ijms21155368>) the results were therefore to be expected. In the same study the similarities with the "DAM" signatures from other mouse model were evaluated and discussed.

We are grateful to the reviewer’s helpful comments to improve our manuscript. We have carefully read the paper by Cougnoux et al. (Single Cell Transcriptome Analysis of Niemann–Pick Disease, Type C1 Cerebella; *International Journal of Molecular Sciences*, 2020, 21(15), 5368). This study has already shown that microglial depletion with PLX3397 treatment did not have a demonstratable effect on either Purkinje cell loss or survival of NPC1 mice. Accordingly, we have revised our manuscript to describe the results in the previous work, as shown below:

Page 26, lines 522-525 in the revised manuscript;
Discussion section

Microglial cells were depleted partially by the gene ablation of *Csf-1* (Fig. 7 and 8) or the oral administration of PLX3397 (Fig. 9 and 10); however, ~~unexpectedly~~, the Purkinje cell

degeneration was not affected in these mouse models. **The results are consistent with a previous report (Cougnoux et al., 2020).**

In the report by Cougnoux et al. (Single Cell Transcriptome Analysis of Niemann–Pick Disease, Type C1 Cerebella; *International Journal of Molecular Sciences*, 2020, 21(15), 5368), the authors demonstrated that a subset of NPC1 microglia can be classified as DAM; the result supported the concept in their previous report that subpopulations of microglia have neurotoxic or neuroprotective activity in a NPC1 mouse model of necroptosis inhibition combined with HP β CD treatment (Cougnoux et al., Necroptosis inhibition as a therapy for Niemann-Pick disease, type C1: Inhibition of RIP kinases and combination therapy with 2-hydroxypropyl-beta-cyclodextrin; *Mol. Genet. Metab.* 2018, 125, 345–350). Accordingly, we have revised our manuscript to describe the previous works, as shown below:

Page 27, lines 547-550 in the revised manuscript;

Discussion section

Cougnoux et al. (2020) have shown that a subset of NPC1 microglia can be classified as DAM, supporting their concept that subpopulations of microglia have neurotoxic or neuroprotective activity in NPC1 mice (Cougnoux et al., 2018b, 2020).

References newly cited in the revised manuscript:

Cougnoux, A., Clifford, S., Salman, A., Ng, S. L., Bertin, J., & Porter, F. D. (2018b).

Necroptosis inhibition as a therapy for Niemann-Pick disease, type C1: Inhibition of RIP kinases and combination therapy with 2-hydroxypropyl- β -cyclodextrin.

Molecular Genetics and Metabolism, 125(4), 345–350.

<https://doi.org/10.1016/J.YMGME.2018.10.009>

Cougnoux, A., Yerger, J. C., Fellmeth, M., Serra-Vinardell, J., Martin, K., Navid, F., Iben, J., Wassif, C. A., Cawley, N. X., & Porter, F. D. (2020). Single cell transcriptome

analysis of niemann–pick disease, type c1 cerebella. *International Journal of Molecular Sciences*, 21(15), 5368. <https://doi.org/10.3390/ijms21155368>

- Newton and coworker (<https://www.ncbi.nlm.nih.gov/pmc/articles/PMC5349795/>) evaluated the effect of FTY720 in NPC. In this study the authors suggested it may be a promising avenue in NPC. Based on your results lymphodepletion would exacerbate the disease. These maybe conflicting observations should be discussed?

We appreciate the reviewer’s excellent comment to improve our manuscript. We agree to the reviewer’s comment that the conflicting observations about a therapeutic potential of FTY720 should be discussed. In the report by Newton et al. (FTY720-fingolimod increases NPC1 and NPC2 expression and reduces cholesterol and sphingolipid accumulation in Niemann-Pick type C mutant fibroblasts; *FASEB Journal*, 2017, 31(4), 1719–1730), they investigated a therapeutic potential of FTY720, a sphingosine 1-phosphate receptor modulator that is frequently used for the treatment of MS. This medicine prevents the egress of lymphocytes from secondary lymphoid organs leading to immune system suppression. Newton et al. demonstrated in the report that treatment of normal mice, mouse NIH 3T3 cells, and NPC1 mutant human fibroblasts with FTY720 enhanced expression of NPC1 and NPC2 and decreased accumulation of cellular cholesterol and glycosphingolipids in an SphK2-dependent manner. These protective effects have been exerted through the activity of phosphorylated FTY720 as an HDAC inhibitor in the nucleus. FTY720 accumulates in the CNS and has several advantages over other HDAC inhibitors in clinical trials, providing a potential opportunity for treatment of the patients with NPC1. However, this study has not investigated the effect of lymphodepletion on neuronal degeneration in NPC1 mutant mice. Since the NPC1 mutant mice used in our study are NPC1-null model in which *Npc1* gene is disrupted by replacement of wild-type genomic sequence with retroposon-like DNA, we speculated that the therapeutic potential of FTY720 should be evaluated using other mutant NPC1 animal model with a single nucleotide substitution. Accordingly, we have revised the description that reviewer pointed

out as shown below. And, we will greatly appreciate the reviewer if he/she could check whether the revised manuscript could discuss the conflicting observations precisely:

Pages 21-22, lines 431-439 in the revised manuscript;

Discussion section

Newton et al. (2017) demonstrated that FTY720/fingolimod, an inhibitor of histone deacetylase, enhanced expression of NPC1 and decreased accumulation of cellular cholesterol in NPC1 mutant human fibroblasts (Newton et al., 2017). This report provided a potential therapeutic opportunity to treat the patients with NPC. However, FTY720/fingolimod acts as a modulator of sphingosine-1-phosphate receptor to prevent the egress of lymphocytes from secondary lymphoid organs and is frequently used for the treatment of MS (Dumitrescu et al., 2023; McGinley & Cohen, 2021). The neuroprotective property of FTY720/fingolimod should be carefully evaluated in NPC1 mutant animals.

References newly cited in the revised manuscript:

- Newton, J., Hait, N. C., Maceyka, M., Colaco, A., Maczys, M., Wassif, C. A., Cougnoux, A., Porter, F. D., Milstien, S., Platt, N., Platt, F. M., & Spiegel, S. (2017). FTY720/fingolimod increases NPC1 and NPC2 expression and reduces cholesterol and sphingolipid accumulation in Niemann-Pick type C mutant fibroblasts. *The FASEB Journal*, *31*(4), 1719–1730. <https://doi.org/10.1096/FJ.201601041R>
- McGinley, M. P., & Cohen, J. A. (2021). Sphingosine 1-phosphate receptor modulators in multiple sclerosis and other conditions. *Lancet*, *398*(10306), 1184–1194. [https://doi.org/10.1016/S0140-6736\(21\)00244-0](https://doi.org/10.1016/S0140-6736(21)00244-0)
- Dumitrescu, L., Papathanasiou, A., Coclitu, C., Garjani, A., Evangelou, N., Constantinescu, C. S., Popescu, B. O., & Tanasescu, R. (2023). An update on the use of sphingosine 1-phosphate receptor modulators for the treatment of relapsing multiple sclerosis. *Expert Opinion on Pharmacotherapy*, *24*(4), 495–509. <https://doi.org/10.1080/14656566.2023.2178898>

- From the genes up regulated in the monocytes and suggested to be involved in the pathology several have already been studied and could be included in the discussion or at least referenced:

(<https://www.ahajournals.org/doi/10.1161/CIRCULATIONAHA.107.701276> ;

<https://www.ncbi.nlm.nih.gov/pmc/articles/PMC3877698/> ;

<https://www.ncbi.nlm.nih.gov/pmc/articles/PMC6935239/>)

We are grateful to the reviewer for his/her helpful comments to improve our manuscript.

Accordingly, we have added the following sentence as reviewer pointed out:

Page 25, lines 501-502 in the revised manuscript;

Discussion section

Similar results have been reported previously (Cologna et al., 2014; Cougnoux et al., 2019; Welch et al., 2007).

References newly cited in the revised manuscript:

Cologna, S. M., Cluzeau, C. V. M., Yanjanin, N. M., Blank, P. S., Dail, M. K., Siebel, S., Toth, C. L., Wassif, C. A., Lieberman, A. P., & Porter, F. D. (2014). Human and mouse neuroinflammation markers in Niemann-Pick disease, type C1. *Journal of Inherited Metabolic Disease*, 37(1), 83–92.

<https://doi.org/10.1007/s10545-013-9610-6>

Cougnoux, Antony, Drummond, R. A., Fellmeth, M., Navid, F., Collar, A. L., Iben, J., Kulkarni, A. B., Pickel, J., Schiffmann, R., Wassif, C. A., Cawley, N. X., Lionakis, M. S., & Porter, F. D. (2019). Unique molecular signature in mucopolipidosis type IV microglia. *Journal of Neuroinflammation*, 16(1), 276.

<https://doi.org/10.1186/s12974-019-1672-4>

Welch, C. L., Sun, Y., Arey, B. J., Lemaitre, V., Sharma, N., Ishibashi, M., Sayers, S., Li, R., Gorelik, A., Pleskac, N., Collins-Fletcher, K., Yasuda, Y., Bromme, D.,

D'Armiento, J. M., Ogletree, M. L., & Tall, A. R. (2007). Spontaneous atherothrombosis and medial degradation in Apoe^{-/-}, Npc1^{-/-} mice. *Circulation*, 116(21), 2444–2452. <https://doi.org/10.1161/CIRCULATIONAHA.107.701276>

- TLR4 was found constitutively activated in NPC (<https://www.jneurosci.org/content/27/8/1879>) it is surprising not to find any signature of this mechanism in your analysis. Do you have any hypothesis regarding this?

We appreciate the reviewer's helpful comments to improve our manuscript. We have carefully read the paper by Suzuki et al. (Endosomal Accumulation of Toll-Like Receptor 4 Causes Constitutive Secretion of Cytokines and Activation of Signal Transducers and Activators of Transcription in Niemann–Pick Disease Type C (NPC) Fibroblasts: A Potential Basis for Glial Cell Activation in the NPC Brain; *Journal of Neuroscience*, 2007, 27(8), 1879–1891). This report demonstrated that human NPC fibroblasts and mouse glial cells constitutively secrete inflammatory cytokines that is partly caused by endosomal accumulation of TLR4. Based on these important results, we speculated that the activation of TLR4 and production of cytokines in the cerebral cells including astrocytes may attract peripheral immune cells, or conversely, may be a response of the cerebral cells to inflammatory stimulus of the infiltrating immune cells. Accordingly, we have revised our manuscript to add the following sentence:

Page 24, lines 485-488 in the revised manuscript;

Discussion section

Constitutive activation of Toll-like receptor 4 and production of inflammatory cytokines in NPC1 cerebral cells, including astrocytes (Suzuki et al., 2007), may also attract peripheral cells, while these changes may potentially be induced by the infiltrating immune cells.

References newly cited in the revised manuscript:

Suzuki, M., Sugimoto, Y., Ohsaki, Y., Ueno, M., Kato, S., Kitamura, Y., Hosokawa, H., Davies, J. P., Ioannou, Y. A., Vanier, M. T., Ohno, K., & Ninomiya, H. (2007). Endosomal accumulation of Toll-like receptor 4 causes constitutive secretion of cytokines and activation of signal transducers and activators of transcription in Niemann-Pick disease type C (NPC) fibroblasts: A potential basis for glial cell activation in the NPC brain. *Journal of Neuroscience*, 27(8), 1879–1891. <https://doi.org/10.1523/JNEUROSCI.5282-06.2007>

- It might me more accurate to describe the rotarod test as a motor coordination test rather than a behavior test.

We appreciate the reviewer’s helpful comments to improve our manuscript. We will agree to the reviewer’s suggestion that it may be more accurate to describe the rotarod test as a motor coordination test rather than a behavior test. Accordingly, we have revised the description that reviewer pointed out as shown below:

Page 30, line 631 in the revised manuscript;

Materials and methods section

Motor coordination test

- The reference for g:Profiler is missing (<https://pubmed.ncbi.nlm.nih.gov/27098042/>)

We appreciate the reviewer’s helpful comments to improve our manuscript. Accordingly, we have revised our manuscript to add the new reference that reviewer pointed out as shown below:

Page 34, lines 717-718 in the revised manuscript;

Materials and methods section

The GO analyses were performed through webtools g:Profiler (<https://biit.cs.ut.ee/gprofiler/gost>; (Reimand et al., 2016))

References newly cited in the revised manuscript:

Reimand, J., Arak, T., Adler, P., Kolberg, L., Reisberg, S., Peterson, H., & Vilo, J. (2016). g:Profiler-a web server for functional interpretation of gene lists (2016 update). *Nucleic Acids Research*, 44(W1), W83–W89. <https://doi.org/10.1093/NAR/GKW199>

- The reference for EdgeR is missing (<https://pubmed.ncbi.nlm.nih.gov/19910308/>)

We appreciate the reviewer’s helpful comments to improve our manuscript. Accordingly, we have revised our manuscript to add the new reference that reviewer pointed out as shown below:

Page 34, lines 715-717 in the revised manuscript;

Materials and methods section

Gene transcripts were quantified by using a tool salmon (Patro et al., 2017) and were analyzed by EdgeR (Robinson et al., 2010).

References newly cited in the revised manuscript:

Robinson, M. D., McCarthy, D. J., & Smyth, G. K. (2010). edgeR: A Bioconductor package for differential expression analysis of digital gene expression data. *Bioinformatics*, 26(1), 139–140. <https://doi.org/10.1093/bioinformatics/btp616>

- Several reports, including in NPC, have shown that Gapdh shouldn't be used as internal control for myeloid cells

(e.g. :<https://bmcimmunol.biomedcentral.com/articles/10.1186/1471-2172-11-21>). Why were the genes expression normalized to Gapdh? This has to be justified or additional

experiments demonstrating that Gapdh has a stable expression between control and mutant cells should be performed or at least evaluated in the transcriptomic data.

We will deeply appreciate the reviewer's excellent comment to improve our manuscript. We agree to the reviewer's suggestion that normalization of gene expression to Gapdh should be justified. We have carefully read the paper by Piehler et al. (Gene expression results in lipopolysaccharide-stimulated monocytes depend significantly on the choice of reference genes; BMC Immunology, 2010, 11, 21). This study demonstrated that Gapdh is not stable upon stimulation with LPS in human peripheral monocytes. The authors have described in the Conclusion section as shown below:

The importance of our findings is highlighted by the fact that a review of the literature on gene expression in LPS-stimulated monocytes of the last years exhibited that the large majority of the published RT-qPCR results were normalized to a single gene, mainly GAPDH or ACTB.

Due to the absence of universal reference genes, however, the state-of-the-art evaluation of reference gene stability has to be documented for each experimental setup and tailored to every activation process.

Accordingly, as the reviewer also pointed out, we have evaluated our transcriptomic data and found no significant difference in the expression level of Gapdh between control and NPC1. And also, we speculated that citation of several references that show RT-qPCR data of NPC1 mice, which are normalized to Gapdh, can be helpful for the readers to evaluate our present data. Accordingly, we have revised our manuscript to provide previous RT-qPCR data in NPC1 mice as shown below:

Page 15, lines 305-308 in the revised manuscript;

Results section

The data are expressed as mean values relative to the housekeeping gene (*gapdh*) that has been used as a reference gene in previous works in NPC1 mice (Cluzeau et al., 2012; Cologna et al., 2014), and they are normalized to the npc1^{+/+}_5wk group.

Responses to the comments by Reviewer #3

Protocol: What is meant by BBB breaching? Blood-brain barrier? Please explain

We are grateful to the reviewer for his/her helpful comments to improve our manuscript. We agree to the reviewer's comment that BBB should be explained precisely. Accordingly, we have revised the description that reviewer pointed out as shown below:

Page 29, line 602 in the revised manuscript

Materials and methods section

Blood-brain barrier breaching

A concern is the illustrations underpinning the FACS and survival data: Morphological evidence is provided for microglia infiltration; but there is not much evidence for lymphocyte infiltration or macrophage infiltration. The demonstration of perivascular cells does not really convincingly show direct interaction between Purkinje cells and cells of the immune system. Please explain

We will appreciate the reviewer's excellent comment to improve our manuscript. According to the reviewer's comment, we have provided a representative picture of gating strategy for FACS analysis to isolate peripheral CD11b-positive monocytes from npc1^{-/-}

and npc1^{+/+} control mice as shown in the revised Supplemental figure 3 A. Accordingly, we have revised our manuscript to provide the new illustration as shown below: (Supplemental figure 3 A-E in the previously submitted manuscript has been revised as Supplemental figure 3 B-F in the revised manuscript)

Revised Supplemental figure 3 A

Page 4, lines 85-88 in the revised Supplemental figure legends;

(A) Gating strategy for FACS analysis to isolate peripheral CD11b-positive monocytes from 7-wk-old female npc1^{-/-} and from npc1^{+/+} control littermates. A representative FACS plot is shown.

According to the reviewer's comment, we have increased the number of mice for survival analysis and provided the p values of log-rank tests. The numbers of mice analyzed for npc1^{-/-}_Ccr2^{+/+} and npc1^{-/-}_Ccr2^{R/R} groups have been increased from (n = 8-13 mice per group) to (n = 12-22 mice per group). Accordingly, we have revised our manuscript to provide the new data as shown below:

Page 81, lines 1441-1442 in the revised manuscript;

Figure legends section

No significant differences were observed between the groups (p = 0.604).

Pages 3-4, lines 62-64 in the revised Supplemental figure legends;

No significant differences were identified between the $npc1^{-/-}$ _Rag1 $^{+/+}$ and the $npc1^{-/-}$ _Rag1 $^{-/-}$ groups in either gender ($p = 0.618$ in female and $p = 0.171$ in male).

Page 4, lines 79-81 in the revised Supplemental figure legends;

Survival time was analyzed for female (F) and male (G) $npc1^{-/-}$ _Ccr2 $^{+/+}$ and $npc1^{-/-}$ _Ccr2-R/R mice (n = 12–22 mice per group).

Page 4, lines 81-83 in the revised Supplemental figure legends;

No significant differences were identified between the $npc1^{-/-}$ _Ccr2 $^{+/+}$ and the $npc1^{-/-}$ _Ccr2-R/R groups in either gender ($p = 0.892$ in female and $p = 0.855$ in male).

We could not detect infiltration of lymphocytes into the CNS by flow cytometry and histological analysis. Several previous studies that are listed below (Cognoux et al. 2018, 2020; Cho et al. 2019) have failed to identify infiltrating immune cells in the cerebellum of NPC1 mice by flow cytometry and single cell transcriptomic analysis.

- 1) Cognoux et al. (Microglia activation in Niemann–Pick disease, type C1 is amenable to therapeutic intervention; Human Molecular Genetics, 2018, Vol. 27, No. 12, 2076–2089)
- 2) Cho et al. (Absence of infiltrating peripheral myeloid cells in the brains of mouse models of lysosomal storage disorders; Journal of Neurochemistry, 2019, 148(5), 625–638)
- 3) Cognoux et al. (Single Cell Transcriptome Analysis of Niemann–Pick Disease, Type C1 Cerebella; International Journal of Molecular Sciences, 2020, 21(15), 5368)

To analyze infiltrating immune cells in the cerebellum, Cognoux et al. (2018) and Cho et al. (2019) performed flow cytometry, and Cognoux et al. (2020) performed single cell transcriptome analysis. As described in the paper by Cognoux et al. (2020), $npc1^{-/-}$ cells may have increased sensitivity to the enzymatic disassociation required to

prepare samples for flow cytometry and single cell transcriptome analysis. We also speculated that a small number of infiltrating cells in NPC1 were difficult to detect when compared with those in experimental autoimmune encephalomyelitis model (Cho et al. 2019) or mucopolidosis type IV model (Cognoux et al. 2018). Moreover, in our present study, the histological detection of infiltrating CD4⁺ lymphocytes and Foxp3-GFP⁺ Treg cells was difficult because the cerebellum of NPC1 mice had a weak and broad autofluorescence. Accordingly, we have revised the description as shown below:

Page 23, lines 456-459 in the revised manuscript;

Discussion section

Previous works failed to detect infiltrating cells in the cerebella of NPC1 mice, possibly due to an increased sensitivity of NPC1 cells to the enzymatic dissociation required for flow cytometry and single cell transcriptome analyses (Cho et al., 2019; Cognoux et al., 2018a, 2020).

References newly cited in the revised manuscript:

- Cho, S. M., Vardi, A., Platt, N., & Futerman, A. H. (2019). Absence of infiltrating peripheral myeloid cells in the brains of mouse models of lysosomal storage disorders. *Journal of Neurochemistry*, *148*(5), 625–638. <https://doi.org/10.1111/jnc.14483>
- Cognoux, A., Yerger, J. C., Fellmeth, M., Serra-Vinardell, J., Martin, K., Navid, F., Iben, J., Wassif, C. A., Cawley, N. X., & Porter, F. D. (2020). Single cell transcriptome analysis of niemann–pick disease, type c1 cerebella. *International Journal of Molecular Sciences*, *21*(15), 5368. <https://doi.org/10.3390/ijms21155368>

We will deeply appreciate the reviewer’s excellent comment to improve our manuscript. We totally agree to the reviewer’s suggestion that the demonstration of perivascular cells does not convincingly show direct interaction between Purkinje cells and immune cells and this important point should be explained.

We newly carried out bone marrow transplantation using $npc1^{-/-}$ _KuO-transgenic mice as a donor. As shown in the revised Supplemental figure 1 S-Z, we found an invasion of the KuO-CD11b-double-positive monocyte-derived macrophages not only into the perivascular zone of the interlobular space, but also into the perivascular zones of molecular layer and Purkinje cell layer of the cerebellum. We speculated that these results strengthen our concept that peripheral immune cells can be involved in the modulation of the neurodegenerative process of NPC1. Accordingly, we have revised our manuscript to provide the new data as shown below:

Revised Supplemental figure 1 S-Z

Pages 1-2, lines 12-27 in the revised Supplemental figure legends;

(S and T) Immunohistochemical analysis for female $npc1^{+/+}$ _BMT (S) and $npc1^{-/-}$ _BMT mice (T) using $npc1^{-/-}$ _KuO-Tg mice as a donor. Sagittal sections of mouse cerebella were immunostained for Purkinje cells (anti-Calbindin, green) and microglia/macrophages (anti-Iba1, red). Note that the KuO^{+} BM-derived cells (blue) were localized at the

interlobular space (indicated by white arrowheads in panel T) and parenchyma of the cerebellum (blue arrowheads in panel T). Scale bar in (S): 500 μm (applicable to panels S and T) (U–Z) Confocal images of the KuO^+ BM-derived cells. Sagittal sections of female mouse cerebella were immunostained for monocytes/macrophages (anti-CD11b, green) and blood vessels (anti-Glut-1, gray). An enlarged images of panel U, W, and Y are indicated in panels V, X, and Z, respectively. Z-stacks were captured at 1.00- μm intervals by using a confocal laser-scanning microscope. Note that the KuO^+ BM-derived cells (red) were CD11b^+ macrophages and localized perivascular zones of the interlobular space (indicated by white arrowhead in panels U and W), the ML (blue arrowhead in panel W), and the PCL of the cerebellum (blue arrowhead in panel Y). Scale bars in (U): 50 μm (applicable to panels W and Y) and (V): 10 μm (applicable to panels X and Z).

Page 10, lines 178-182 in the revised manuscript;

Results section

When the BM cell transplantation was performed using $\text{npc1}^{-/-}$ KuO-Tg mice as a donor, we observed an invasion of the KuO-CD11b -double-positive monocyte-derived macrophages into the perivascular zones of the interlobular space, the molecular layer (ML), and the Purkinje cell layer (PCL) of the cerebellum in both female and male mice (Fig. S1 S–Z).

Page 13, line 256 in the revised manuscript;

Results section

may have infiltrated the CNS (Fig. 2 L–O and Fig. S1 S–Z)

Page 23, lines 453-456 in the revised manuscript;

Discussion section

This finding suggests that macrophages infiltrating into the perivascular zones of the interlobular space (Fig. 2 L–O and Fig. S1 S–Z), the ML, and the PCL of the cerebellum (Fig. S1 S–Z) can promote Purkinje cell degeneration.

Page 24, lines 477-480 in the revised manuscript;

Discussion section

We speculated that the BM-derived macrophages found in the cerebellar perivascular zones (Fig. 2 L–O and Fig. S1 S–Z) may possess distinct properties from the so-called “perivascular macrophages,” and may be involved in neuronal degeneration in a disease-specific manner.

Page 32, line 675 in the revised manuscript;

Materials and methods section

the length of the PCL

Page 32, line 678 in the revised manuscript;

Materials and methods section

the areas of the ML

The following sentence was removed.

Page 24, line 490-491 in the revised manuscript;

Discussion section

Moreover, it may be of critical importance to elucidate the distinct localization and fate of the infiltrating macrophages deriving from *npc1*^{-/-} BM cells.

The comments on the blood-brain barrier in the supplementary are not sufficient. There must be a better provision of the criteria for BBB breaching and the broad staining by Evans Blue is almost too good. The animals must be very sick if this is true BBB breaching; does this lead to labeling of Purkinje cells? Does this breaching lead to extravasation and presence of immunoglobulins in Purkinje cells?

We will appreciate the reviewer's excellent comment to improve our manuscript. We will totally agree to the reviewer's suggestion that a better provision of the criteria for BBB breaching including the broad staining of extravasated dye is necessary. As shown in the revised Supplemental figure 1 G' and H', we have provided a whole brain picture that shows extravasation of sulfo-NHS-biotin in several brain areas. Accordingly, we have revised our manuscript to provide the new data as shown below:

(Supplemental figure 1 C' and D' in the previously submitted manuscript has been revised as Supplemental figure 1 E' and F' in the revised manuscript)

Revised Supplemental figure 1 G' and H'

Pages 2-3, lines 39-45 in the revised Supplemental figure legends;

(E'–H') The 7-wk-old *npc1*^{+/+} and *npc1*^{-/-} mice were transcardially perfused with sulfo-NHS-biotin in PBS, followed by paraformaldehyde (n = 3 each). Nuclei were visualized by DAPI (blue). Note that sulfo-NHS-biotin is extravasated to the ML of the cerebellum **(F')** and **several brain areas (H')** in *npc1*^{-/-} mice, but not in *npc1*^{+/+} mice **(E')**

and G'). The result suggests a breaching of blood-brain-barrier. Scale bars in (E'): 200 μm (applicable to panels E' and F') and in (G'): 1 mm (applicable to panels G' and H').

According to the reviewer's comment that the labeling of Purkinje cells with extravasated dye should be investigated, we have performed immunostaining for Calbindin in the sections of NPC1 mice injected with Evans blue dye. As shown in the revised Supplemental figure 1 C' and D', we observed the extravasation of Evans blue in the ML and the PCL and the labeling of potentially degenerating Purkinje cells (indicated by white arrowheads in panel D').

And also, according to the reviewer's comment, we have performed immunostaining for mouse immunoglobulin G (IgG) in the sections of 7- and 10-wk-old NPC1 and control mice (n = 5–6 mice per group) using Block Ace (Yukijirushi-Nyugyo Co, Sapporo, Japan) to prevent non-specific binding of IgG to the sections. However, we could not detect any positive staining that show extravasation and presence of IgG in Purkinje cells.

Accordingly, we have revised our manuscript to provide the new data from Evans blue experiment as shown below:

Revised Supplemental figure 1 C' and D'

Page 2, lines 33-39 in the revised Supplemental figure legends;

(C' and D') Sagittal sections of female mouse cerebella were immunostained for Purkinje cells (anti-Calbindin, green). Note that Evans blue is extravasated to the ML and the PCL of the cerebellum in *npc1*^{-/-} mice (D'), but not in *npc1*^{+/+} mice (C'). This finding suggests labeling of degenerating Purkinje cells with Evans blue (indicated by white arrowheads in panel D') and breaching of the BBB. Scale bar in (C'): 100 μ m (applicable to panels C' and D').

By submitting the manuscript to Life Science Alliance, I understand that the work described here has not been submitted elsewhere for publication in whole or in part, and all the authors listed have approved the manuscript. I have read and abided by the statement of ethical standards for manuscripts submitted to Life Science Alliance.

Please assess our improved article for publication in Life Science Alliance.

Sincerely yours,

Toru Yasuda, PhD,
National Center for Child Health and Development
Department of Human Genetics
2-10-1 Okura
Setagaya-ku
Tokyo 157-8535
Japan
Phone: +81 (3) 5494-7035
E-mail: yasuda-t@ncchd.go.jp

June 9, 2023

RE: Life Science Alliance Manuscript #LSA-2022-01881-TR

Dr. Toru Yasuda
National Center For Child Health and Development
2-1-1 Okura
Setagaya
Tokyo 157-8535
Japan

Dear Dr. Yasuda,

Thank you for submitting your revised manuscript entitled "Peripheral immune system modulates Purkinje cell degeneration in Niemann-Pick disease type C1". We would be happy to publish your paper in Life Science Alliance pending final revisions necessary to meet our formatting guidelines.

- please address Reviewer 2's remaining minor comment
- please upload your main and supplementary figures as single files
- please add ORCID ID for the corresponding author--you should have received instructions on how to do so
- please add the Twitter handle of your host institute/organization as well as your own or/and one of the authors in our system
- please use the [10 author names et al.] format in your references (i.e., limit the author names to the first 10)
- please add an Author Contributions section to your main manuscript text
- please add your supplementary figure legends to the main manuscript text after the legends for the main figures
- please revise the figure legends for Figure 10 and ensure that the figure panels are introduced alphabetically corresponding to the panels in the figure.
- please add a conflict of interest statement to your main manuscript text
- please add callouts for Figures 2K, 3B, and 4B to your main manuscript text
- please be sure to insert all callouts for figures S1, S2 and S3

A. FINAL FILES:

B. MANUSCRIPT ORGANIZATION AND FORMATTING:

Sincerely,

Reviewer #2 (Comments to the Authors (Required)):

The authors studied the importance of immune cells in the rare lysosomal storage disease Niemann-Pick type C1. The authors repeated previous finding from other groups regarding myeloid cells and shed important new lights on the relevance of T cells in the disease.

The authors performed an in depth study using both pharmacological and genetic tools to identify the weight of multiple immune cells on the disease progression. In addition the authors used extensive immunohistological and molecular techniques which results support the conclusions.

The authors have addressed my previous concern regarding the manuscript.

While I believe the manuscript is suitable for publication I have a minor comment:

Gene nomenclature should be followed: Gapdh instead of gapdh

Reviewer #3 (Comments to the Authors (Required)):

I think the authors have replied adequately in their revision and my recommendation is: Accept

June 14, 2023

RE: Life Science Alliance Manuscript #LSA-2022-01881-TRR

Dr. Toru Yasuda
National Center For Child Health and Development
2-10-1 Okura
Setagaya
Tokyo 157-8535
Japan

Dear Dr. Yasuda,

Thank you for submitting your Research Article entitled "Peripheral immune system modulates Purkinje cell degeneration in Niemann-Pick disease type C1". It is a pleasure to let you know that your manuscript is now accepted for publication in Life Science Alliance. Congratulations on this interesting work.

DISTRIBUTION OF MATERIALS:

Again, congratulations on a very nice paper. I hope you found the review process to be constructive and are pleased with how the manuscript was handled editorially. We look forward to future exciting submissions from your lab.

Sincerely,
